

# Opposite spectral properties of Rossby waves during weak and strong stratospheric polar vortex events

Michael Schutte[1], Daniela I.V. Domeisen[2,3], and Jacopo Riboldi[1,3]

[1]Department of Earth Sciences, Uppsala University, Uppsala, Sweden
[2]Institute of Earth Surface Dynamics, Université de Lausanne, Lausanne, Switzerland
[3]Institute for Atmospheric and Climate Science, ETH Zurich, Zurich, Switzerland

**Correspondence:** Michael Schutte (misc1907@student.uu.se)

**Abstract.** In this study we provide a systematic characterization of Rossby wave activity during the 25 sudden stratospheric warming (SSW) and 31 strong polar vortex (SPV) events that occurred in the period 1979-2021, identifying the specific tropospheric and stratospheric waves displaying anomalous behaviour during such events. Space-time spectral analysis is applied to ERA5 data for this purpose, so that both the wavenumber and the zonal phase speed of the waves can be assessed. We find that SSW events are associated with a reduction in the phase speed of Rossby waves, first in the stratosphere and then in the troposphere; SPV events are tied to a concomitant increase of phase speed across vertical levels. Phase speed anomalies become significant around the event and persist for 2-3 weeks afterwards. Changes of Rossby wave properties in the stratosphere during SSW and SPV events are dominated by changes in the background flow, with a systematic reduction or increase, respectively, in eastward propagation of the waves across most wavenumbers. In the troposphere, on the other hand, the effect of the background flow is also complemented by changes in wave properties, with a shift towards higher wavenumbers during SSW events and towards lower wavenumbers for SPV events. The opposite response between SSW and SPV events is also visible in the meridional heat and momentum flux co-spectra, which highlight from a novel perspective the connection between stratospheric Rossby waves and upward propagation of waves.



## 1 Introduction

Due to the absence of local baroclinic instability and non-adiabatic forcing (e.g., topography, latent heat release, land-sea contrast), Rossby waves in the stratosphere reflect to a large extent the upward propagation of waves from below (e.g., Hartmann, 1985). Periods of enhanced upward propagation and wave breaking tend to be followed by a weakening of the stratospheric westerlies (Matsuno, 1971; Sjoberg and Birner, 2012; Polvani and Waugh, 2004; Reichler and Jucker, 2022): episodes when the zonally averaged flow reverses from westerly to easterly, together with a rapid increase in temperature in the polar strato-

sphere, are termed sudden stratospheric warming events (SSWs; see the review by Baldwin et al., 2021). On the other hand, a weaker than usual upward wave propagation can lead to a strengthening of the polar vortex (Limpasuvan et al., 2005), i.e., to a strong polar vortex (SPV) event. The anomalous flow associated with these opposite, extreme stratospheric states can affect the tropospheric circulation and surface weather (Baldwin and Dunkerton, 2001; Hinssen et al., 2011; Kodera et al., 2013; Davini et al., 2014; Afargan-Gerstman et al., 2020; Domeisen et al., 2020c; White et al., 2020). Previous research has particularly

focused on SSW events because of their linkage to extreme weather events (e.g., Domeisen and Butler, 2020; Finke et al., 2023) and periods of higher than usual extended-range and seasonal predictability (e.g., Domeisen et al., 2020b; Zhang et al., 2022), while SPVs have received comparatively less attention, although these events have also been linked to increased surface predictability (Tripathi et al., 2015).

Tropospheric and stratospheric Rossby waves can thus actively affect, but in turn their propagation is also affected by the state

of the stratospheric background flow: the structure and strength of the three-dimensional background flow in which the waves propagate can cause them to break (e.g., in case of easterly flow; Charney and Drazin, 1961) or waves can be reflected back towards the surface (Harnik and Lindzen, 2001). Both stationary (i.e., waves with no zonal phase speed) and zonally traveling waves can propagate upward under certain background flow conditions, and both can contribute to the genesis of SSW events (Domeisen and Plumb, 2012). While such events are usually studied in terms of stationary wave forcing, the propagation of

planetary-scale waves with enhanced positive (i.e., eastward-traveling) phase speed precedes some SSW events (Madden and Labitzke, 1981; Rhodes et al., 2021), which according to Charney and Drazin (1961) can enhance their propagation and thereby may contribute to the increase in wave forcing in the polar vortex region ahead of SSWs (Domeisen et al., 2018). However, a systematic investigation of the role of zonal phase propagation and spectral behavior of Rossby waves for different types of stratospheric extreme events is still pending.

The impact of SSW and SPV events onto the tropospheric circulation is most often analyzed in terms of circulation indices (such as the Arctic Oscillation or the Northern Annular Mode, e.g., Baldwin and Dunkerton, 2001; Thompson et al., 2006) of zonally averaged quantities (e.g., Hall et al., 2021), or in terms of changes in the frequency of weather regimes (e.g., Charlton-Perez et al., 2018; Domeisen et al., 2020c; Hall et al., 2023). These approaches, however, do not explicitly consider the modification of Rossby wave activity accompanying the events themselves. Such an analysis is complicated by the fact

that Rossby waves span across different temporal and spatial scales (e.g., traveling vs quasi-stationary waves, planetary- vs synoptic-scale waves), often interacting among themselves in a nonlinear fashion. For instance, it is not yet clear whether planetary-scale or synoptic-scale tropospheric waves are the main modulator of the surface impact of SSW events (Song and



Robinson, 2004; Smith and Scott, 2016), as the two types of waves are interconnected in complex feedbacks (Domeisen et al., 2013; White et al., 2020).

This complex wave behavior can be investigated by means of space-time spectral analysis, which allows for a decomposition of a time-varying wave pattern (e.g., the one visualized by an Hovmöller diagram) into a basis of harmonics with different horizontal scales and phase speeds. Such a technique has already been employed to compare Rossby wave properties across different data sets (Dell'Aquila et al., 2005) and periods (Riboldi et al., 2020; Sussman et al., 2020), and to investigate circumpolar Rossby wave patterns during boreal winter (Riboldi et al., 2022). Furthermore, the same technique can be used to

compute space-time co-spectra, which provide an estimate of the contribution of each harmonic to the observed covariance between wind and temperature waves (for the eddy heat flux; e.g., Randel and Held, 1991) and between the zonal and meridional wind (for the eddy momentum flux). These two quantities are related, respectively, to the upward propagation of baroclinic waves and to their breaking, and can thus be insightful to study Rossby wave modifications before and after extreme states of the stratospheric polar vortex.

Here we investigate a few questions related to the properties of Rossby waves during SSW and SPV events: which Rossby wave harmonics are enhanced, or weakened, during SSW and SPV events? Which ones contribute the most to heat and momentum fluxes? Is there a preference for higher or lower phase speeds than usual before and after such events? The manuscript is structured as follows: The employed spectral decomposition and the procedure for significance assessment are described in Sec. 2. The seasonally averaged spectra for the extended winter period (November to March) are introduced in Sec. 3 to

verify that they provide a realistic representation of the tropospheric and stratospheric circulation. The impact of SSW and SPV events onto stratospheric and tropospheric Rossby wave spectra is then described in Sec. 4, while the corresponding heat flux and momentum flux co-spectra are discussed in Sec. 5 and Sec. 6, respectively. The results are summarized and discussed in Sec. 7.

## 2  Data and Methods

The analysis is based on ERA5 reanalysis data (Hersbach et al., 2020) of meridional wind and temperature on pressure levels, interpolated to a spatial resolution of 0.75°×0.75° and with 6-hourly temporal resolution between March 1979 and February 2021. As the stratospheric polar vortex is most active during November to March (NDJFM), only data for this time window will be used in the following analysis.

### 2.1  Definition of SPV and SSW events

Following the definition by Charlton and Polvani (2007), an SSW event occurs if the zonally averaged 10 hPa wind speeds at 60°N reverse direction from westerly to easterly. In accordance with this definition, Butler (2020) catalogued 26 SSWs recorded within the ERA-Interim reanalysis data set (Dee et al., 2011) for the Northern Hemisphere in the period from 1979 to 2020 in line with the methodology from Butler et al. (2017). Even though that set of events is based on ERA-Interim, it matches well the SSW dates in the ERA5 reanalysis (Afargan-Gerstman, personal communication). Excluding the SSW event





in February 1979, as it lies outside of the here studied time period, the remaining 25 events constitute our set of SSWs in the following analysis.

As for SSW events, we employ zonally averaged wind speed at $10\,\mathrm{hPa}$ and $60°N$ to identify SPV events. Even though an optimal, physically-based threshold to define SPV events has not been determined yet, we here expand on the approach by Oehrlein et al. (2020) and choose the value of $48\,\mathrm{m\,s^{-1}}$ as an optimal (although rather arbitrary) threshold. Starting from the

absolute maximum value, groups of neighboring days above the threshold are merged together and considered as candidate events. To be considered as SPV events, candidate events must be separated by at least 20 days and last for a minimum of 5 days: this leads to a total of 31 SPV events. The date of maximum zonally averaged wind is chosen as the one corresponding to the SPV central date, rather than the one corresponding to the first day where the wind threshold was exceeded. This choice differs from other definitions (e.g., Oehrlein et al., 2020; Domeisen et al., 2020a) and is motivated by the fact that the time

span between the day when the threshold is first crossed and the wind maximum can be very large for SPV events (16.5 days on average, but more than 28 days for 7 of the 31 events identified for this study; see Table A1). The resulting superposition of different stages of the SPV event at different lead times can result in a systematic blurring of the spectral response, in particular for long-lasting, intense SPV events. This is not the case for SSW events, where event start and event peak tend to occur closely in time (3.8 days on average; see Table A1).

## 2.2   Space-time Rossby wave spectra and co-spectra

Rossby wave spectra and co-spectra are obtained from a double Fourier transform, in space and time, and displayed in harmonics with wavenumber $n$ and phase speed $c_p$ (as in, e.g., Randel and Held, 1991; Domeisen et al., 2018; Riboldi et al., 2022). We follow the methodology by Riboldi et al. (2022), with the only difference being that here raw data are used to compute the spectra instead of anomalies with respect to the seasonal cycle (de-seasonalisation is performed on the spectra instead).

The two-dimensional fields of zonal wind $U$, meridional wind $V$, and temperature $T$ at $250\,\mathrm{hPa}$ are decomposed on each latitude circle $\phi$ as a linear superposition of zonally propagating waves following the formula

$$X(\lambda,t;\phi) = \sum_{j=-N_T/2}^{N_T/2} \sum_{n=-N_L/2}^{N_L/2} \hat{X}(n,\omega_j;\phi)\, e^{i(n\lambda-\omega_j t)} \tag{1}$$

where $\hat{X}(n,\omega_j;\phi)$ correspond to the complex Fourier coefficients of each variable ($U$, $V$, and $T$, respectively), $\lambda$ is longitude in radians (from 0 to $2\pi$) and $t$ is time, $N_L = 720$ is the number of grid points along a given latitude circle and $\omega_j = 2\pi j/N_T$

is the angular frequency, with $N_T = 244$ six-hourly time steps for a total of 61 days. As in Riboldi et al. (2022), the decomposition was performed considering each day as the center of a sliding 61-day time window centered at 12UTC, with a double cosine tapering applied to the first and last 12 days of each window. The periodograms corresponding to the meridional wind spectra are $\mathrm{P}(n,\omega_j;\phi) = 2\mathfrak{Re}(\hat{V}\hat{V}^*)$. The ones corresponding to the heat flux co-spectra are $\mathrm{C}_{\overline{V'T'}}(n,\omega_j;\phi) = 2\mathfrak{Re}(\hat{V}\hat{T}^*)$, while momentum flux co-spectra are $\mathrm{C}_{\overline{U'V'}}(n,\omega_j;\phi) = 2\,|\mathfrak{Re}(\hat{U}\hat{V}^*)|$, where $\hat{V}^*$, $\hat{T}^*$ are the complex conjugates of the corre-

sponding coefficients. The absolute value in the definition of the momentum flux co-spectrum indicates that no distinction is made between cyclonic and anticyclonic wave breaking to avoid cancellation when averaging across different latitudes. Each





periodogram is smoothed 10 times in the frequency direction using a three-point window, as in Wheeler and Kiladis (1999). The interpolation of space/time periodograms from the frequency to the phase speed domain is performed separately for each latitude circle, following Randel and Held (1991). The approach consists in the interpolation of the periodogram along lines

of constant phase speed $c_p = \omega/k$, followed by a re-scaling, and is detailed in the Supplement to Riboldi et al. (2022). The spectra $S_{\overline{V'V'}}(n, c_p)$ and co-spectra $C_{\overline{V'T'}}(n, c_p)$, $C_{\overline{U'V'}}(n, c_p)$ result from the average of the interpolated periodograms across the considered latitude bands (35°N-75°N for the troposphere, 45°N-75°N in the stratosphere), and are attributed to the central day of each 61-day time window. The underlying seasonal cycle is computed for each calendar day and then smoothed using a 30-day running window, and is used to de-seasonalize spectra and co-spectra.

Most studies pre-select by means of spectral or harmonic analysis waves with "planetary" wavenumbers $n$ =1-3 when studying the stratospheric circulation. Even though the criterion by Charney and Drazin (1961) would in principle allow to neglect a-priori the contribution from higher wavenumbers, or from waves with negative phase speeds, we chose here to retain the entirety of the Rossby wave spectrum (note that no explicit filtering is applied to compute the spectra). This can prove insightful to better understand the complex connection between troposphere, where Rossby wave activity is split across several

wavenumbers, and stratosphere (e.g., by considering barotropic modes that would not be allowed by Charney-Drazin's criterion as in Esler and Scott, 2005).

The interpolation of the spectra to the phase speed domain is based on the general definition $c_p = \omega/k$, which in the case of Rossby waves involves both the contribution of the background flow and of the waves themselves. This fact can confound, in principle, the effect onto $c_p$ of changes in the large-scale flow in which the waves propagate (e.g., the transition to easterlies

resulting from an SSW event) and of the changes in the characteristics of the waves. This ambiguity is part of the broader problem of unequivocally separating Rossby waves of finite amplitude and scale from the unperturbed background flow (discussed, among others, by Wirth and Polster, 2021). To deal with this limitation while interpreting the results, we postulate the following "rule-of-thumb": We assume that the background flow contribution equally affects to first order all wavenumbers by shifting the spectra towards lower/higher phase speeds than climatology. To second order, it is clear that according to Charney

and Drazin (1961), even for their assumption of a strongly simplified background flow, different waves are affected differently depending on their wave number and zonal phase speed. On the other hand, changes in the shape of the waves should be visible as shifts towards higher/lower wavenumbers than climatology.

## 2.3 Spectrally-derived metrics

Two integral metrics can be obtained from the space-time spectra $S_{\overline{V'V'}}$ to summarize the overall magnitude of wave activity

and the direction of wave propagation.

The integral of spectral energy density across all harmonics in the $S(n, c_p)$ spectra is called integrated spectral power (ISP), defined as

$$\text{ISP} = \sum_{n=1}^{15} \sum_{c_p=-30}^{30} S(n, c_p) \tag{2}$$

and is used here as an overall estimate of Rossby wave activity in a given time window.





The second metric is an estimate of the hemispherically-averaged Rossby wave phase speed $\overline{c_p}$ across the resolved harmonics, defined in Riboldi et al. (2020) as a weighted mean across each spectrum:

$$\overline{c_p} = \frac{\sum_{n=1}^{15} \sum_{c_p=-30}^{30} S(n, c_p) \cdot c_p}{\sum_{n=1}^{15} \sum_{c_p=-30}^{30} S(n, c_p)} = \frac{1}{\text{ISP}} \sum_{n=1}^{15} \sum_{c_p=-30}^{30} S(n, c_p) \cdot c_p \tag{3}$$

Here, the phase speed associated with each $(n, c_p)$ harmonic is weighted by the associated spectral energy density $S_{\overline{V'V'}}$. In this way, the harmonics that are "active" in the considered time period would contribute more than others to the global value of $\overline{c_p}$.

## 2.4 Significance assessment

The significance of deviations from the climatological mean is addressed using a bootstrapping approach to compute confidence intervals from the daily NDJFM data (Efron and Tibshirani, 1994). Random samples with the same size as the original sample of the event are drawn –with repetition– from the original data for 1000 times to get a reliable distribution of the test samples: from this distribution one obtains confidence levels, and we deem as nonsignificant values that fall in the central 99% of the re-sampled statistic.

With a similar method one can assess which wavenumber-phase speed harmonics deviate significantly from their climatological mean during SSWs and SPVs. At first, a series of 10 consecutive days is obtained, e.g., for 10 to 20 days after the event. If then the mean of the 10 days lies outside the confidence intervals from the climatology, the anomaly is considered significant. The confidence intervals from the climatology are computed by re-sampling. At first, one draws samples of 10 consecutive days as often as the number of single events to calculate the mean of this sample. This process is repeated 1000 times to obtain a reliable distribution of the mean values. From this distribution, confidence levels are obtained such that 99% of the values lie within the confidence interval. Periods around SSW and SPV events are not excluded. The same procedure is applied for the 20 day-windows in the analysis of co-spectra.

## 3 Spectral characteristics of Northern Hemispheric Rossby waves during extended winter

To better contextualize the results for SSW and SPV events, we start here by briefly discussing the average meridional wind spectra, and the heat and momentum flux co-spectra obtained by averaging all the spectra attributed to each day during NDJFM.





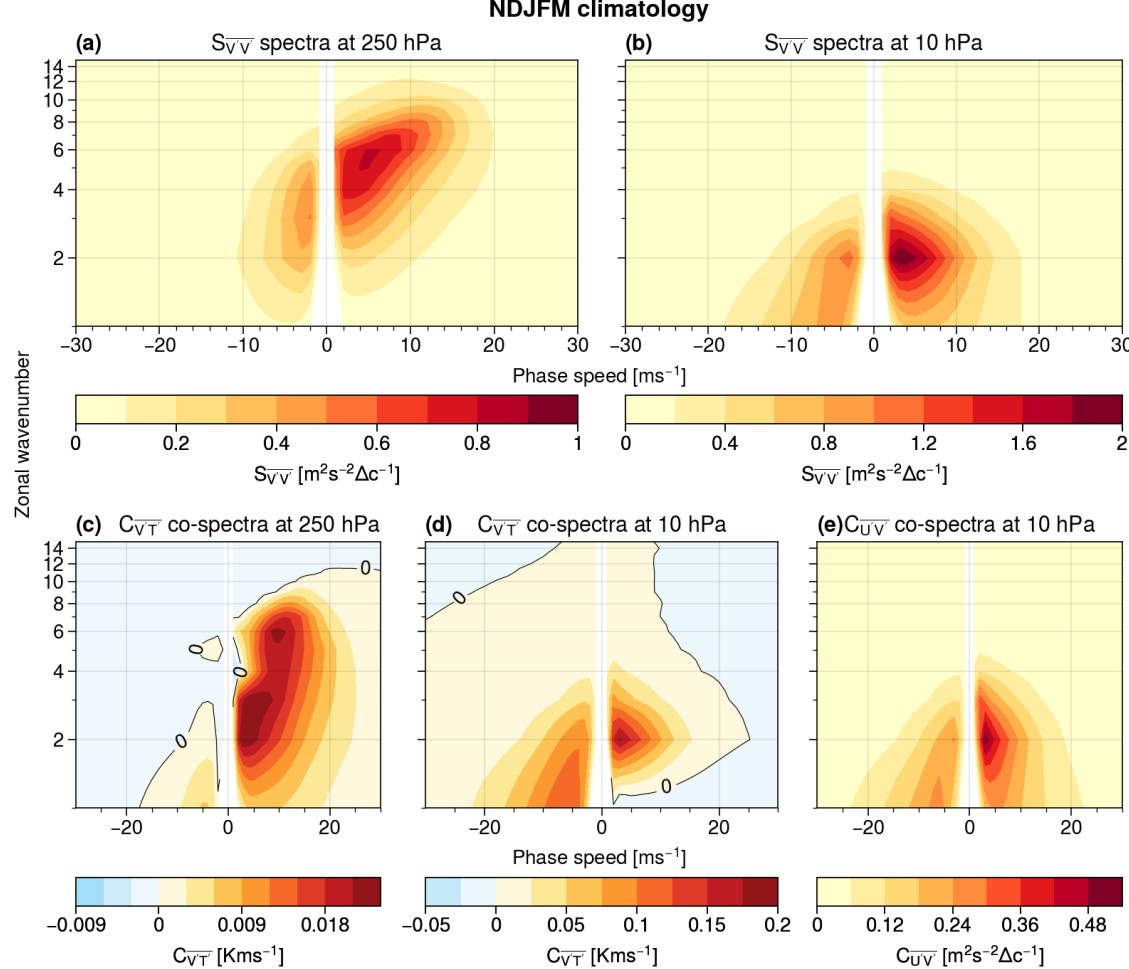

**Figure 1.** Seasonal mean spectral power $S_{\overline{V'V'}}$ of meridional wind V at 250 hPa (a) and 10 hPa (b) for NDJFM, as well as for heat flux co-spectra $C_{\overline{V'T'}}$ at 250 hPa (c) and 10 hPa (d) and momentum flux co-sepctra $C_{\overline{U'V'}}$ at 10 hPa (e). Note the different color scale between the different panels, and that values are multiplied by $n$ to compensate for the non-constant density of points (due to the logarithmic scale used for the y-axis).

At 250 hPa, the highest values of spectral power are found for $n = 4-7$ and phase speeds between $2\,\mathrm{m\,s^{-1}}$ and $8\,\mathrm{m\,s^{-1}}$ (Fig. 1a; see also Riboldi et al., 2022). Spectra of stratospheric Rossby waves at 10 hPa, on the other hand, exhibit most wave energy at low wavenumbers ($n = 1-4$): maximum values exceeding $1.6\ \mathrm{m^2 s^{-2}\Delta c^{-1}}$ (approximately twice as much as the 250 hPa maximum) are found for harmonics with $n = 2$ and at positive phase speeds (Fig. 1b). The reduced spectral power at high wavenumbers is due to the hindered upward propagation of high-wavenumber Rossby waves from the troposphere, consistent with the Charney-Drazin criterion (Charney and Drazin, 1961). Eastward-propagating harmonics have generally higher




power than westward-propagating ones, due to the prevalent westerly winds at both levels, but in certain periods westward-
propagating waves can become more dominant (Riboldi et al., 2020).

The space-time co-spectra of meridional eddy heat transport ($\overline{V'T'}$) indicate which harmonics contribute to the covariance between waves in meridional wind and temperature, with positive values corresponding to poleward heat transport. In the troposphere, most of the heat transport is due to eastward-propagating waves (Fig. 1c). The near-zero values for westward-propagating waves, on the other hand, are indicative of waves without a clear baroclinic structure, i.e., with little or no phase
tilt with height (Mechoso and Hartmann, 1982). Two relative maxima are visible in the average heat flux co-spectra: one corresponds to heat transport by high-wavenumber ($n = 5-7$), rapidly propagating ($c_p = 7-13 \, \mathrm{m\,s^{-1}}$) waves, while the other to heat transport by slow-moving, low-wavenumber waves ($n = 2-3$). These separate maxima were already noticed by Randel and Held (1991) as a feature exclusive to Northern Hemispheric winter, while the contribution of rapid transients dominates in other seasons and in the Southern hemisphere.

Given that meridional heat flux is closely related to the upward propagation of wave energy (Edmon et al., 1980), heat flux co-spectra allow us to discern vertical wave propagation from the troposphere to the stratosphere. For instance, we see that only tropospheric waves corresponding to the heat flux maximum at low wavenumbers seem able to propagate into the stratosphere, while heat flux by rapid transients does not correspond to a signature in the stratosphere for the same range of harmonics (Figs. 1b,d). In the rest of this work, we will use $250 \, \mathrm{hPa}$ co-spectra (Fig. 1c) to diagnose which waves can "depart"
from the troposphere and $10 \, \mathrm{hPa}$ co-spectra (Fig. 1d) to diagnose which waves actually "reach" the level of the stratospheric polar vortex. We note here that also the tropopause and the lower stratosphere (at heights around $100 \, \mathrm{hPa}$) can act as a source of upward-propagating Rossby waves: the exact physical mechanisms behind that source are, however, still a matter of active research (Boljka and Birner, 2020).

Upward-propagating waves from the upper troposphere/lower stratosphere can perturb irreversibly the state of the polar
vortex when they undergo nonlinear wave breaking (Baldwin et al., 2021). Momentum flux co-spectra $C_{\overline{U'V'}}$ indicate which waves contribute the most to momentum fluxes: once more, we observe the signature of low-wavenumber waves (Fig. 1e), that correspond to the same range of harmonics as the vertically propagating ones from below (Fig. 1f). This indicates that, on average, such waves are the most important contributors to the variability in meridional momentum flux.



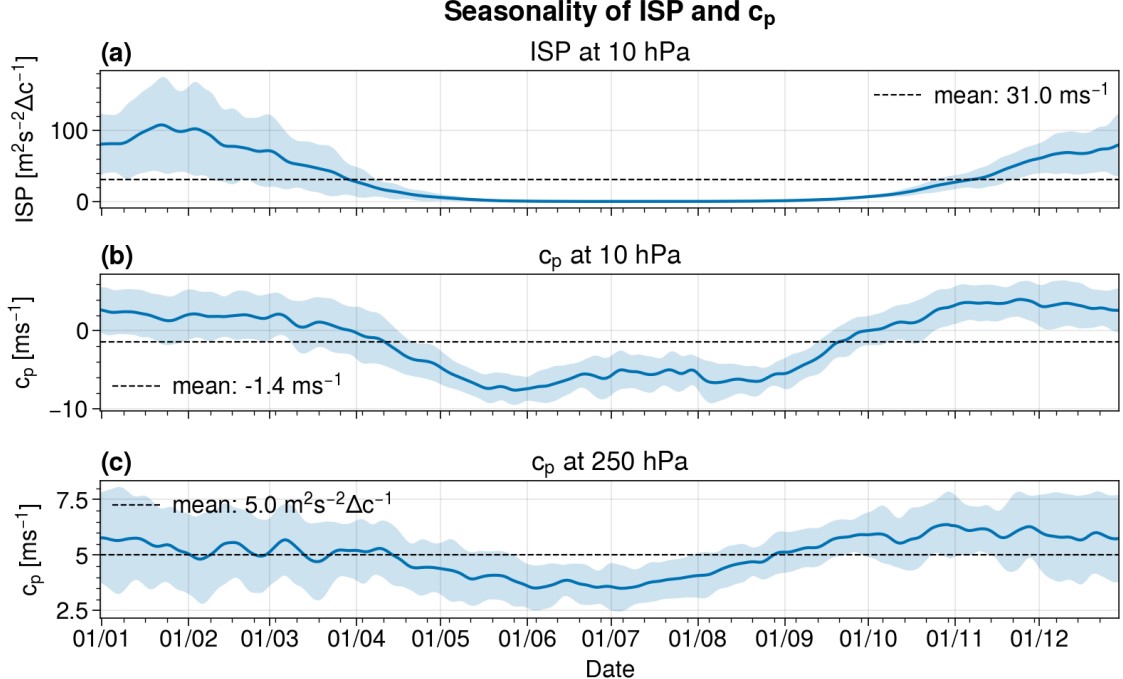

**Figure 2.** Seasonal cycle of ISP at 10 hPa (a), as well as phase speed at 10 hPa (b) and at 250 hPa (c). Blue lines depict the daily average, and blue shaded areas mark $\pm 1$ standard deviation around the daily mean. The dashed line shows the mean over the whole time period from 1979 to 2020.

The importance of vertical wave propagation becomes particularly visible if we consider the seasonal cycle of ISP across
the whole year (Fig. 2a). Rossby wave activity is two orders of magnitude lower in summer than in winter, approaching 0
$\text{m}^2\text{s}^{-2}\Delta c^{-1}$ between May and October, as the dominant easterly flow in the stratosphere inhibits upward wave propagation
(again, according to Charney and Drazin, 1961). At the same time, phase speeds at 10 hPa turn from positive, between October
and April, to negative during summer, following the reversal of the background stratospheric winds from westerly to easterly
during the warm season (Fig. 2b; remind that $\overline{c_p}$ values during summer correspond to a virtually non existent Rossby wave
activity). Phase speeds at 250 hPa, instead, are positive throughout the whole year with lower phase speeds occurring between
April and September (Fig. 2c; see also Riboldi et al., 2020, for more details).

## 4  Rossby wave spectra during SSW and SPV events

After having verified that space-time spectra can provide a realistic representation of Rossby wave activity in the stratosphere
and in the troposphere, the focus shifts to the identified sets of 25 SSW and 31 SPV events. Composite analysis is here used
to highlight systematic changes in Rossby wave shape and propagation before, during and after such extreme states of the
stratospheric polar vortex. Relative anomalies of spectral power with respect to the climatologies of Fig. 1 will be shown to



highlight significant deviations more easily. For spectra, relative anomalies are computed by dividing the value of the anomaly by the NDJFM (November to March) seasonal mean for each wavenumber-phase speed harmonic. Hence, a value of -1 indicates a reduction of spectral power by 100% with respect to the seasonal mean, whereas +1 indicates a twice as high value. For co-spectra, which are not necessarily positively defined, standardized anomalies (with respect to the standard deviation computed over the whole NDJFM period) will be shown.

## 4.1 SSW events

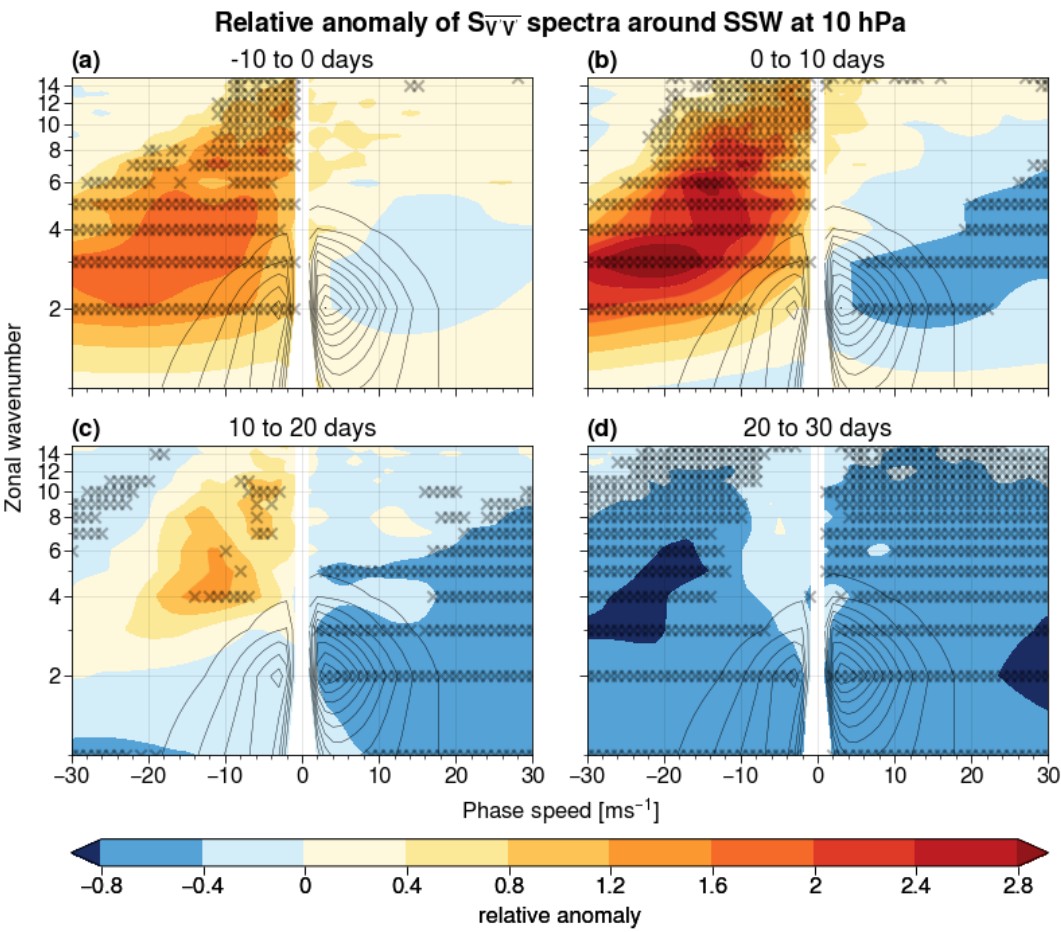

**Figure 3.** Relative anomalies of spectral power $S_{\overline{V'V'}}$ at $10\,\mathrm{hPa}$ (with respect to the mean spectrum) averaged over 10-day time intervals around SSW events (shaded). Subplot (a) shows spectra for the period from 10 days prior to the event to the event start, subplot (b) shows spectra between the event start and 10 days after the event start, (c) for 10 to 20 days, and (d) for 20 to 30 days after the event. Wavenumber-phase speed pairs marked with an $\times$ exceed the 0.5th or 99.5th percentile of the re-sampled distribution. Black contour lines show the NDJFM climatology ranging from $0.2\,\mathrm{m^2\,s^{-2}\,\Delta c^{-1}}$ to $1.8\,\mathrm{m^2\,s^{-2}\,\Delta c^{-1}}$ in steps of $0.2\,\mathrm{m^2\,s^{-2}\,\Delta c^{-1}}$ as in Figure 1b.





Rossby wave activity at 10 hPa exhibits a positive anomaly for Rossby waves with negative phase speeds in the days preceding the SSW (Fig. 3a). This positive anomaly is strongest immediately following the onset of the SSW and is contrasted by a

negative anomaly for eastward-propagating harmonics (Fig. 3b). Even if the most prominent anomalies are located outside the climatological range (with average spectral power smaller than $0.2\,\mathrm{m^2\,s^{-2}\,\Delta c^{-1}}$), deviations of the order of 40-80% are visible for $n = 2-3$. The response to SSW events can be interpreted as an overall shift in Rossby wave activity towards a more stationary pattern: given that most wavenumbers are affected at once in the same way, we speculate that it might be related to an overall weakening of the background flow associated with the disruption of the stratospheric polar vortex.

The negative anomaly already visible at positive phase speeds in the 10 days preceding the SSW continues to increase in magnitude in the weeks following the onset of the SSW (Fig. 3c). Approximately 20 to 30 days following the onset of the SSW we notice widespread negative anomalies, with spectral power reduced on average by more than 40% in comparison to climatology, across a majority of wavenumber-phase speed harmonics (even for harmonics with a climatologically high spectral power, see Fig. 3d). Similarly to summer conditions, this global suppression of Rossby wave activity is due to the

impossibility of vertical propagation in easterly flow for tropospheric Rossby waves, as the broad region of easterlies induced by the SSW reaches the lower stratosphere in the weeks following the start of the event.

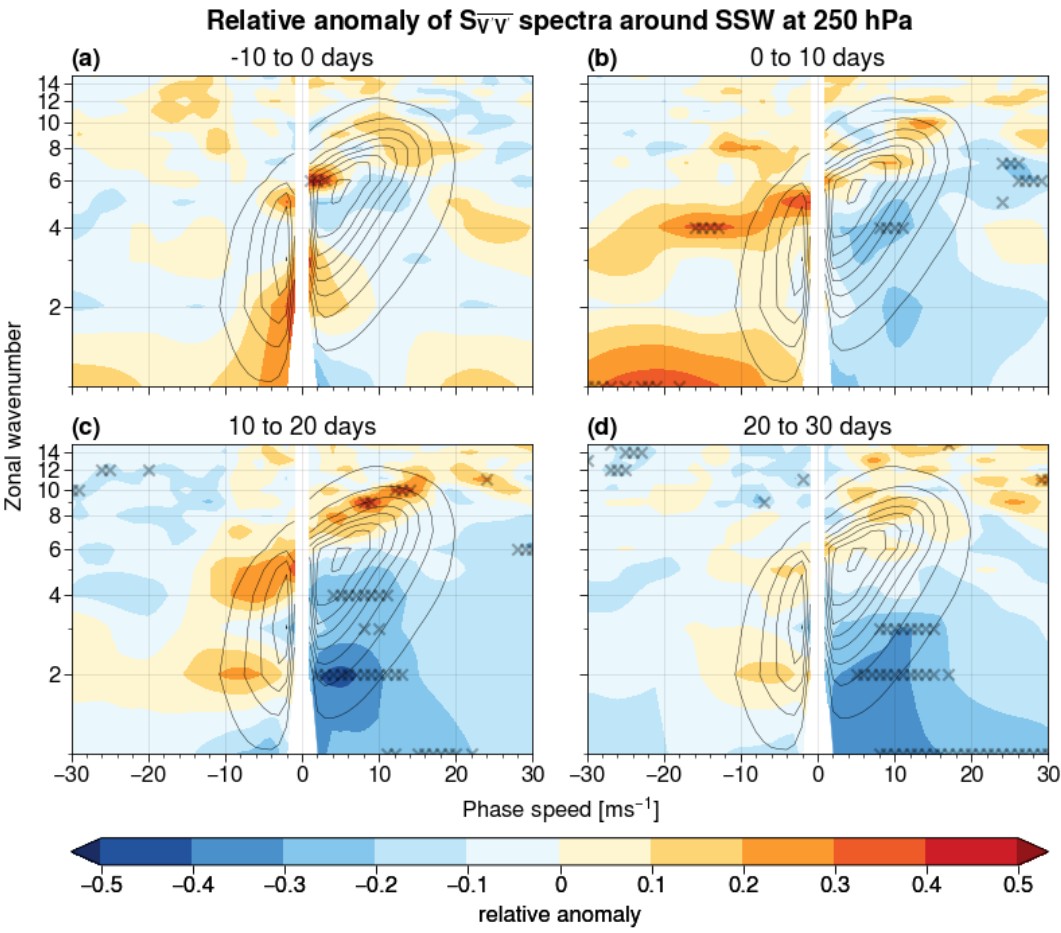

**Figure 4.** Relative anomalies of spectral power $S_{\overline{v'v'}}$ compared to the mean spectrum at 250 hPa, averaged over 10-day time intervals, around SSW events (shaded). Subplot (a) shows spectra for 10 days prior to the event to the event start, subplot (b) shows spectra between the event start and 10 days after the event start, (c) for 10 to 20 days, and (d) for 20 to 30 days after the event. Wavenumber-phase speed pairs marked with an × exceed the 0.5th or 99.5th percentile of the re-sampled distribution. Black contour lines show the NDJFM climatology ranging from $0.1\,\mathrm{m^2\,s^{-2}\,\Delta c^{-1}}$ to $0.8\,\mathrm{m^2\,s^{-2}\,\Delta c^{-1}}$ in steps of $0.1\,\mathrm{m^2\,s^{-2}\,\Delta c^{-1}}$ as in Fig. 1a.

Shifting now the focus on the tropospheric response, no significant anomalies in Rossby wave spectra at 250 hPa can be seen in the days preceding the SSW (Fig. 4a). In particular, no significant anomalies in spectral power at this level are visible more than ten days before the SSW (not shown). A tendency towards lower phase speeds becomes apparent as the SSW unfolds, with

negative anomalies for low-wavenumber ($n = 1 - 4$), eastward-propagating waves 2-4 weeks after event onset and (mostly not significant) positive anomalies for harmonics with negative phase speeds (Figs. 4c-d). This indicates a higher persistence of the large-scale tropospheric flow after SSW events. Previous research has shown that this behavior is particularly evident in the North Atlantic region, where the negative phase of the North Atlantic Oscillation pattern has a higher persistence after SSW events as compared to climatology (Charlton-Perez et al., 2018; Domeisen, 2019).



In addition to the wave deceleration, a shift towards higher wavenumbers is visible for eastward propagating harmonics: this might be related to the effect of synoptic eddies, which have been shown to be important for the tropospheric response to SSW events (Domeisen et al., 2013; White et al., 2020). Alternatively, or in addition, it might also indicate an equatorward shift of eastward-propagating, baroclinic eddies (which have a fixed horizontal scale and, thus, would project on higher wavenumbers at low latitudes). Regardless of the exact explanation, the tropospheric response to SSW events appears to be more complex

than the stratospheric one, as it features modifications in the types of waves involved rather than a simple weakening of the background flow.

## 4.2   SPV events

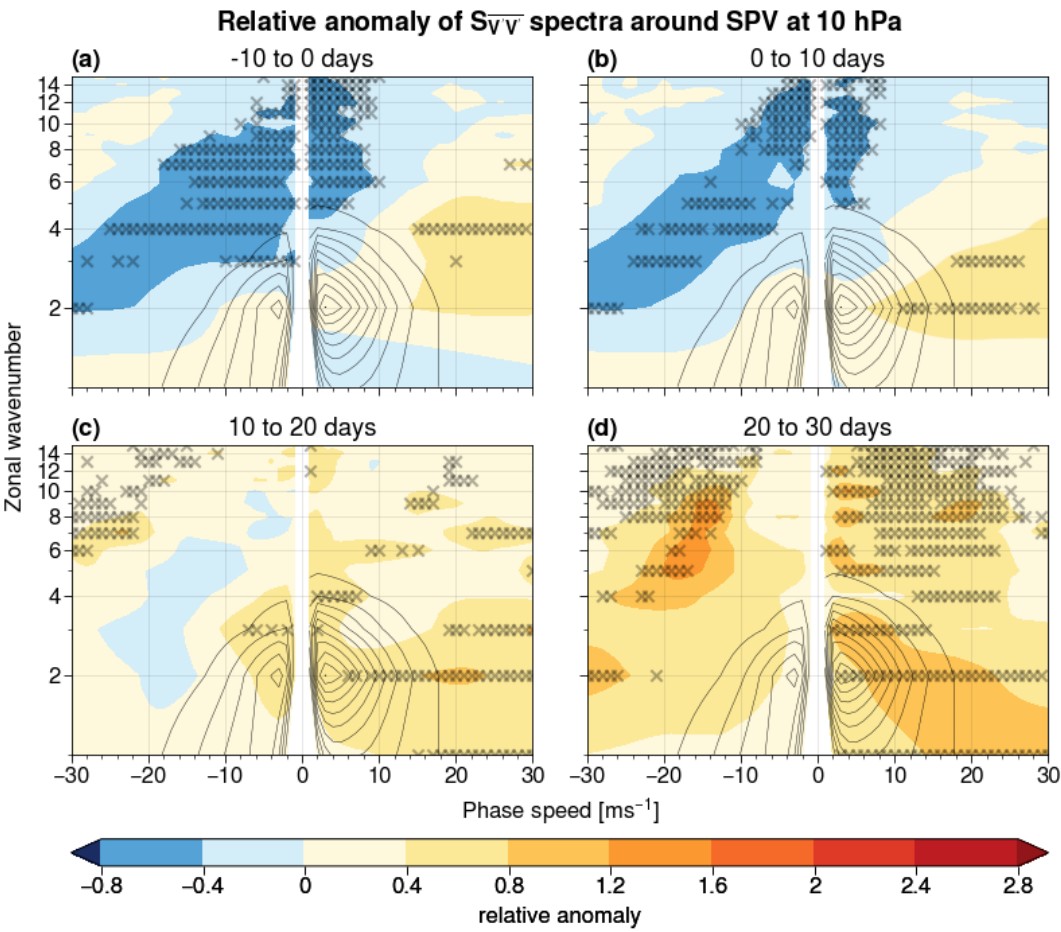

**Figure 5.** Same as Figure 3, but for SPV events.

Stratospheric Rossby wave activity during SPV events exhibits a behavior that is opposite to that observed during SSWs. The presence of negative anomalies for harmonics with negative phase speeds in the initial stages of SPV events, and of positive





anomalies at positive phase speeds, indicates an overall shift in Rossby wave activity towards higher phase speeds than usual (Figs. 5a,b). The positive anomalies strengthen and extend to the whole spectrum in the 3-4 weeks after the event, reaching in particular eastward-propagating harmonics characterized by high climatological spectral power (Figs. 5c,d). This effect can be attributed to the stronger zonal wind speed during SPV events, which serves to advect waves eastward at a faster rate, and to the facilitated upward propagation of eastward-propagating Rossby waves in the enhanced westerly flow (Domeisen et al.,

2018). Times earlier than 10 days before SPVs display on average no significant anomalies (not shown).

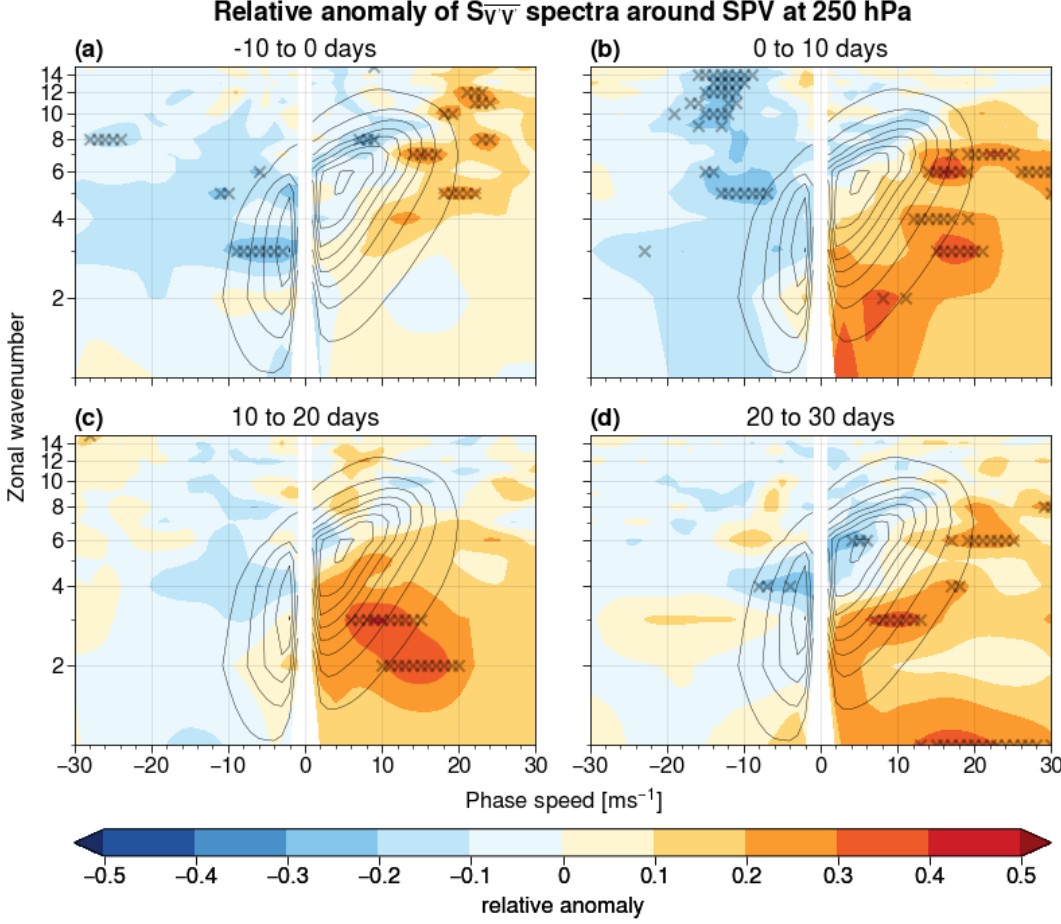

**Figure 6.** Same as Figure 4, but for SPV events.

As in the stratosphere, the observed anomalies of spectral power at 250 hPa during SPV events are to a large extent the opposite of SSWs (Fig. 6) and also do not exhibit significance earlier than 10 days just preceding the event (not shown). Interestingly, positive anomalies for eastward-propagating waves occur in the upper troposphere already in the days prior to the SPV (Fig. 6a). Such anomalies, visible also for harmonics with climatologically high spectral power, are contrasted by

small, mostly not significant negative anomalies at low phase speeds (Fig. 6b-d). This pattern is tied to a general shift in



spectral power towards higher positive phase speeds, indicating a more rapid zonal propagation of Rossby waves, and to an enhancement of eastward-propagating, low-wavenumber harmonics.

As for SSWs, the anomalies associated with SPV events in the troposphere are generally weaker compared to those observed in the stratosphere. However, the antipodal anomalies in eastward-propagating, low-wavenumber harmonics between SSW and SPV events, as well as the opposing overall shifts towards low/high phase speeds, allow us to pinpoint more precisely the impact of extreme states of the polar vortex onto the large-scale circulation in the troposphere. In particular, the eastward-propagating range of low-wavenumber, planetary-scale waves is weakened in the weeks following SSW events, and conversely it is enhanced following SPV events: thus, this range of tropospheric harmonics seems to be particularly "sensitive" to extreme states of the stratospheric polar vortex. This result complements our knowledge of the role played by planetary waves in modulating the tropospheric impact of SSW and SPV events (Song and Robinson, 2004; Domeisen et al., 2013; Smith and Scott, 2016).

### 4.3 Phase speed and ISP

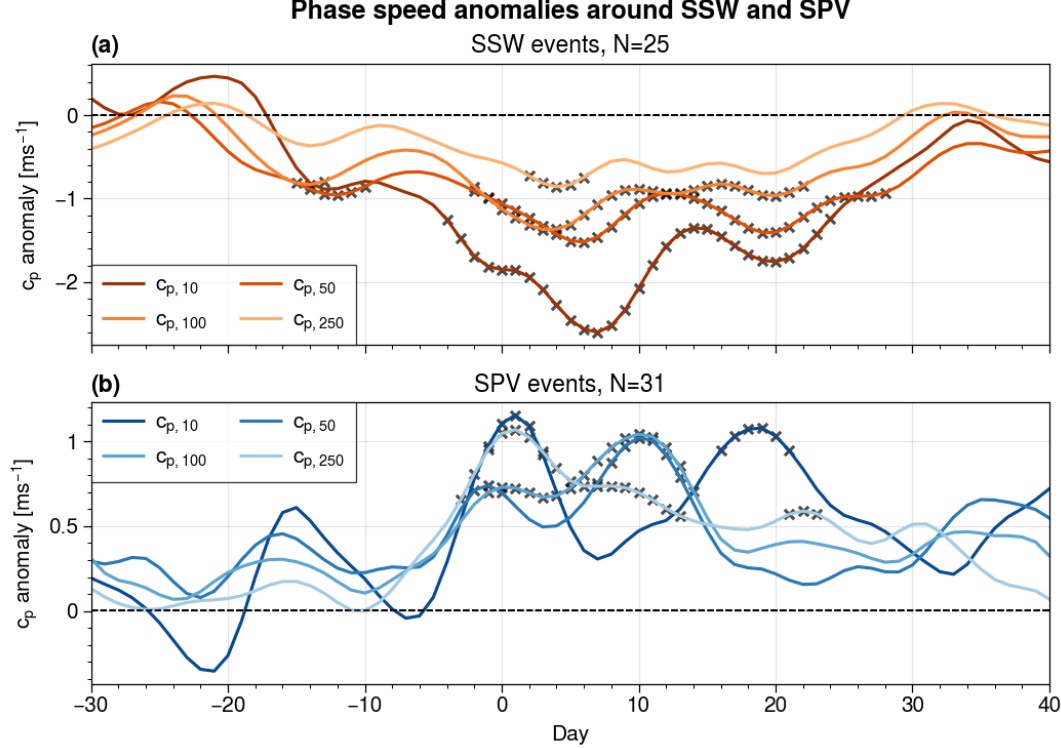

**Figure 7.** Time series of daily averages around SSW (a) and SPV (b) events for phase speed anomaly with respect to the NDJFM seasonal cycle at each level. Points marked with an × lie outside of the 99% confidence interval with respect to the seasonal cycle.





Hemispherically averaged phase speed $\overline{c_p}$ shows a decrease at all pressure levels around SSW events, first in the upper and then in the lower stratosphere (Fig. 7a). Significantly lower values of $\overline{c_p}$ are visible as early as 15 days prior to the events in the upper stratosphere (10-50 hPa), and become significant in the lower stratosphere (100 hPa) and upper troposphere (250 hPa) around SSW onset. In particular, phase speeds at 250 hPa shows a significant decrease in the 3 to 7 days after the SSW (and remain negative, although marginally insignificant, until 21 days), while negative $\overline{c_p}$ anomalies persist in the stratosphere for more than three weeks. The significant decrease in phase speed can be explained by the breakdown of the polar vortex during SSWs, which is by definition connected to a weakening of the zonal background wind.

SPV events show an opposite behaviour to SSW in a number of aspects. Firstly, the expected $\overline{c_p}$ increase becomes significant only around event onset, and this happens roughly at the same time across vertical levels (Fig. 7b). Secondly, tropospheric phase speed anomalies appear more pronounced and persistent than for SSW events. Thirdly, the temporal evolution of $\overline{c_p}$ across levels features a notable temporal variability, in particular at 10 hPa. Two relative maxima are found at that level, one just after event peak and another 15-20 days afterwards. The intermediate minimum between such maxima corresponds, on the other hand, to a $\overline{c_p}$ maximum at lower stratospheric levels (50 hPa and 100 hPa). Such a maximum appears 6 to 10 days after the SPV event peak, thus following the $\overline{c_p}$ maxima observed above (at 10 hPa) and below (at 250 hPa). As anomalies in $\overline{c_p}$ become significant first at 250 hPa and then at 10 hPa, we speculate that a particularly rapid and zonally oriented flow in the troposphere can be beneficial to SPV events, as it would correspond to an absence of amplified waves with a tendency to vertical propagation and breaking. This complex interplay between upper and lower levels might also explain why SPV events tend to have a longer duration (and potentially, multiple peaks) than SSW events.

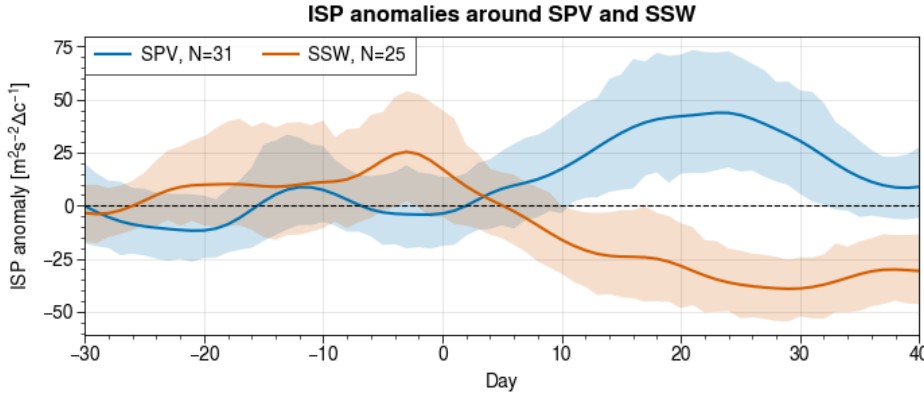

**Figure 8.** Time series of daily averages around SSW (orange) and SPV (blue) events for ISP anomaly with respect to the NDJFM seasonal cycle at 10 hPa. Shaded areas depict the 99% confidence interval with respect to the daily mean.

The evolution of ISP at 10 hPa is consistent with what can be inferred from the spectra: stratospheric Rossby wave activity features higher than usual values in the weeks prior to SSW events, in line with an enhanced upward wave propagation and breaking, followed by a steady decrease after SSW onset (orange line in Fig. 8). Negative anomalies reach $50 \mathrm{m^2 s^{-2}} \Delta \mathrm{c}^{-1}$, in stark contrast with the seasonal average of approximately $70 \ \mathrm{m^2 s^{-2}} \Delta \mathrm{c}^{-1}$, and indicate a significant reduction in Rossby





295  wave activity. This drastic decrease is, as discussed in the previous section, due to a weakened Rossby wave propagation from lower levels. Before SPV events, the ISP oscillates around the climatological average (blue line in Fig. 8). After the event has started, it consistently increases towards strongly positive anomalies, which are comparable in magnitude to SSW events. A peak is reached between 15 and 25 days after the SPV, whereas ISP remains on a plateau of negative anomalies even 30 days after an SSW event (Fig. 8).

300  **5  Heat flux co-spectra during SSW and SPV events**

The analysis of heat flux co-spectra allows us to decompose vertical Rossby wave propagation into contributions from different harmonics, and will be employed to link tropospheric or lower stratospheric precursors to extreme stratospheric events. Significant differences between SSW and SPVs emerge in the stratosphere as well as in the troposphere.



## 5.1 Stratospheric heat flux co-spectra



**Figure 9.** Standardized anomalies (shaded) of heat flux co-spectra $C_{\overline{V'T'}}$ with respect to the NDJFM standard deviation at $10\,\text{hPa}$, averaged over 20-day time intervals around the 25 SSW (a to c) and 31 SPV events (d to f). Wavenumber-phase speed harmonics marked with an $\times$ exceed the 0.5th or 99.5th percentile of the re-sampled distribution. Black contour lines show the NDJFM climatology ranging from -0.025 $\text{K}\,\text{m}\,\text{s}^{-1}\,\Delta\,\text{c}^{-1}$ to 0.175 $\text{K}\,\text{m}\,\text{s}^{-1}\,\Delta\,\text{c}^{-1}$ in steps of 0.025 $\text{K}\,\text{m}\,\text{s}^{-1}\,\Delta\,\text{c}^{-1}$ as in Fig. 1d.

Prior to SSWs, the overall pattern exhibits an enhanced (although insignificant) contribution to meridional heat flux by westward-propagating waves (Fig. 9a). This is opposite to what is observed in the weeks preceding SPV events, where significantly negative covariance anomalies are present for quasi-stationary waves at all wavenumbers (Fig. 9d). Those anomalies are well visible even if harmonics with $c_p = 0\,\text{ms}^{-1}$ cannot be resolved in this setup, as this would require a time interval of infinite duration (see the Supplement of Riboldi et al., 2022, for a detailed explanation). This pinpoints the importance of a suppression of quasi-stationary ($c_p \approx 0$), upward-propagating Rossby waves before SPV events.





The differences between the events become even more prominent for the days around the onset, with harmonics at negative phase speeds responsible for a stronger than usual upward-propagation of Rossby waves during SSW events (Fig. 9b). In particular, the preferential upward propagation of waves with a tendency to westward propagation seems to be particularly important to force the stratospheric polar vortex into a SSW. As the shift towards lower phase speeds is visible across most 315 wavenumbers, the "rule-of-thumb" proposed in Sec. 2 indicates the contribution of the weakened westerly flow during SSW events. On the other hand, the negative anomalies in $C_{\overline{V'T'}}$ across most harmonics observed in the weeks after the event depict the suppression of upward Rossby wave propagation to the stratosphere discussed earlier (Fig. 9c).

The heat flux co-spectra for SPVs again show a mostly opposite behaviour to SSWs (Fig. 9e-f). Negative anomalies persist and become even more present for negative and around-zero phase speeds as the peak of the SPV events is reached, indicating a 320 reduced upward propagation of slow-moving waves (Fig. 9e). This suppressed upward Rossby wave propagation could indicate a decoupling of the stratospheric polar vortex from the troposphere in the SPV buildup, allowing a more efficient spin up of the vortex itself. Following the SPVs, on the other hand, co-spectra tend to be higher than on seasonal average, most notably for eastward-propagating $n = 1$ waves (Fig. 9f).

Slow-moving and westward-propagating Rossby waves significantly contribute to meridional heat flux during SSW, even at 325 high wavenumbers usually not considered so relevant for the stratospheric circulation. On the other hand, the contribution of the same harmonics is significantly smaller than usual during SPV events. Such symmetric anomalies between SSW and SPV events point to a potential role played by those waves in the evolution of SSW and SPV events. The shift towards lower phase speeds during SSW events is also consistent with the the behavior of Rossby wave spectra in the stratosphere (Fig. 3) and at 250 hPa (Fig. 4).





**5.2 Tropospheric heat flux co-spectra**

**Standardized anomaly of C$_{\overline{VT}}$ co-spectra at 250 hPa around SSW and SPV**

**Figure 10.** Standardized anomalies of heat flux co-spectra C$_{\overline{V'T'}}$ compared to the NDJFM standard deviation at 250 hPa, averaged over 20-day time intervals around 25 SSW events (a to c) and over 31 SPV events (d to f). Wavenumber-phase speed harmonics marked with an × exceed the 0.5th or 99.5th percentile of the re-sampled distribution. Black contour lines show the NDJFM climatology ranging from -0.006 $\mathrm{K\,m\,s^{-1}\,\Delta\,c^{-1}}$ to 0.021 $\mathrm{K\,m\,s^{-1}\,\Delta\,c^{-1}}$ in steps of 0.003 $\mathrm{K\,m\,s^{-1}\,\Delta\,c^{-1}}$ as in Figure 1c.

As for the previous results, also the anomalies in the heat flux co-spectra are smaller in magnitude in the troposphere than in the stratosphere due to additional factors influencing the behaviour of the tropospheric jet stream. Prior to SSWs, the contribution to heat flux at 250 hPa by westward-propagating harmonics is lower than the seasonal average (Fig. 10 a). This pattern is opposite to what is observed at 10 hPa, even though these negative anomalies are mostly located outside the range of climatological variability in the co-spectra. The lack of a significant difference between SSW and SPV across most harmonics prior to the

event can be seen as a sign of case-to-case variability in the SSW set (e.g., split vs displacement cases), or can point to the role of lower-stratospheric Rossby wave sources (see, e.g, de la Cámara et al., 2019).





The enhanced contribution of such harmonics to heat flux appears again around the onset of the SSW event, although hardly any anomaly is significant: this is likely indicative of the substantial case-to-case variability between single events, and more

generally of the higher complexity of the tropospheric circulation with respect to the stratospheric one (Fig. 10b). The upper-tropospheric response to the SSW becomes more visible 10 to 30 days after the event onset, as indicated by a significant suppression (around 30%) of the heat flux contribution by eastward-propagating, planetary wave 1 and 2 (Fig. 10 c). This weaker-than-usual contribution to heat flux for such harmonics compounds with the effect of the SSW-related easterlies in the stratosphere, resulting in an even more reduced capability for the troposphere to affect the state of the stratospheric polar vortex

after an SSW event.

Hardly any significant anomalies are present in the heat flux co-spectra in the weeks prior to SPVs (Fig. 10d). Consistently with the spectra, a general shift towards positive phase speeds is observed in the co-spectra as the SPV event unfolds: this is reflected by the negative anomalies for quasi-stationary waves for $n \geq 3$ (Fig. 10e). This shift persists also after the event, associated with a significant weakening of heat flux contribution by quasi-stationary, sub-planetary waves ($n = 3 - 6$), while

positive anomalies remain insignificant (Fig. 10f). The symmetry of the response between SSW and SPV events is noteworthy, as it corresponds to the same range of harmonics in $S_{\overline{V'V'}}$ discussed in Sec. 4.

## 6 Momentum flux co-spectra during SSW and SPV events

The analysis of momentum flux co-spectra allows us to decompose the contributions to Rossby wave breaking in the stratosphere between different harmonics.





## 6.1 Stratospheric momentum flux co-spectra

**Figure 11.** Standardized anomalies (shaded) of absolute momentum flux co-spectra $C_{\overline{U'V'}}$ with respect to the NDJFM standard deviation at 10 hPa, averaged over 20-day time intervals around the 25 SSW (a to c) and 31 SPV events (d to f). Wavenumber-phase speed harmonics marked with an × exceed the 0.5th or 99.5th percentile of the re-sampled distribution. Black contour lines show the NDJFM climatology ranging from $0.06\,\mathrm{m}^2\,\mathrm{s}^{-2}\,\Delta\mathrm{c}^{-1}$ to $0.48\,\mathrm{m}^2\,\mathrm{s}^{-2}\,\Delta\mathrm{c}^{-1}$ in steps of $0.06\,\mathrm{m}^2\,\mathrm{s}^{-2}\,\Delta\mathrm{c}^{-1}$ as in Figure 1e.

No significant anomaly is present for $C_{\overline{U'V'}}$ at 10 hPa in the 30 to 10 days before SSW and SPV events (Figs. 11a,b), indicating only minor changes in Rossby wave breaking before the event occurrence (Fig. 11d). The comparatively stronger signal in heat flux (Figs. 9a,d) than in momentum flux (Figs. 11a,d) suggests that the enhanced upward propagation of Rossby waves does not immediately reflect into Rossby wave breaking at 10 hPa.

During the onset of SSW events, enhanced wave breaking is visible for westward-propagating waves with $n \geq 1$ and for eastward propagating wave-1 and wave-4 (Fig. 11b). This is to be expected, given that the deposition of easterly momentum by breaking waves is one of the main drivers of polar vortex deceleration during SSW events. However, it is interesting to observe





the symmetric reduction in Rossby wave breaking visible around the peak of SPV events for the same range of westward-propagating and quasi-stationary waves with $n \geq 5$ (Fig. 11e). At the same time, SPV events feature a positive momentum flux

contribution for fast, eastward-propagating waves with $n = 2, 4$, which also contrasts with the weaker (although not significant) Rossby wave breaking in the same range of harmonics at the onset of SSW events (Fig. 11b). Such a symmetric pattern of anomalies can be attributed at a first order to the different background zonal flow advecting Rossby waves, stronger for SPV events and weaker for SSW events. This pattern is also very similar to the heat flux co-spectra for the same time range (Figs. 9b,e), indicating that changes in Rossby wave breaking are tied to the different phase speed of upward propagating waves

during SSW and SPV events.

As upward wave propagation is suppressed after SSW events, Rossby wave breaking is also reduced by more than 40% across most harmonics, specifically across westward-propagating waves 1 and 2 and for all eastward-propagating components of the spectrum (Fig. 11c). On the other hand, there is a significant increase in the momentum flux contribution of westward-propagating waves 1 and 2 and most eastward-propagating waves after SPVs, which point to enhanced Rossby wave breaking

following SPVs (Fig. 11f). Such an enhancement of wave breaking after SPV events can in principle set the stage for a deceleration of the polar vortex (Domeisen et al., 2018; Wu et al., 2022): however, we note that SSWs are associated with the preferential breaking of westward-propagating, quasi-stationary Rossby waves (Fig. 11b), while the increase following SPV mostly involves eastward-propagating waves. A similar consideration holds for the heat flux co-spectra (Fig. 9b), indicating that the phase speed of the upward-propagating waves is tied to their effect onto the stratospheric polar vortex.

## 7  Conclusions and Outlook

Extreme states of the stratospheric polar vortex, such as SSW and SPV events, are associated with generally opposite behaviour in terms of stratospheric and tropospheric Rossby wave activity. Space-time spectral analysis allows us to visualize the evolution of such events at the level of the different types of Rossby waves involved, highlighting at the same time changes in shape and propagation of the waves.

**Spectra and co-spectra** The most significant anomalies in the spectra and co-spectra of Rossby waves are visible in the stratosphere, and feature a shift towards a weaker (for SSWs) or stronger (for SPVs) eastward propagation for Rossby waves across most wavenumbers. The reduced eastward propagation of Rossby waves observed during SSW events is concomitant to a substantial increase in the upward propagation and breaking of westward-propagating and quasi-stationary waves, both at high and low wavenumbers. Afterwards, the easterly flow in the stratosphere induced by the SSW compounds with changes

in tropospheric wave activity to suppress vertical wave propagation in the weeks following the event, leading to a drastic reduction of stratospheric Rossby wave activity and breaking. The onset of SPV events, on the other hand, is preceded by a reduced upward propagation of quasi-stationary waves from lower levels, symmetrically to SSW events, and is associated with a more rapid eastward propagation of Rossby waves.

**Phase speed** The occurrence of SSW events is associated with an overall lower zonal phase speed of Rossby waves than

usual, both in the stratosphere and in the troposphere. The deceleration of Rossby waves starts in the stratosphere already two




weeks before the SSW onset, as has e.g. been observed for the upper stratosphere for the 2009 SSW event (Rhodes et al., 2021), while its tropospheric signature appears only after the start of the event and remains visible in the three weeks following the flow reversal at 10 hPa. This reduction in eastward propagation of tropospheric Rossby waves is consistent with previous research discussing a higher-than-usual frequency and persistence of a negative state of the North Atlantic Oscillation in the
North Atlantic region (Charlton-Perez et al., 2018; Domeisen, 2019). On the other hand, SPV events are related to a shift of the Rossby wave pattern towards higher phase speeds, in the stratosphere as well as in the troposphere, that persists during the following weeks.

**Background flow vs wave contribution** The definition of phase speed employed in this study conflates both the effects of the background flow and of the waves. To disentangle them, we here propose a "rule-of-thumb" based on the pattern of
anomalies in the spectra and co-spectra to distinguish between the contribution of the background flow (i.e., a shift in the phase speed direction across several wavenumbers) and the contribution of the waves (i.e., a shift in wavenumber across multiple phase speed). Using this first order distinction, the stratospheric response to SSW and SPV events is dominated by changes in the background flow. This might be simply due to the fact that such events are defined from the zonally averaged zonal wind at 60°N, which can be seen as a proxy of the background flow itself. On the other hand, the tropospheric response to
SSW and SPV events is more complex, with a shift in the properties of waves superimposed on a change in background flow. In particular, significant negative (for SSW events) and positive (for SPV events) anomalies in spectral energy density appear for eastward-propagating, low-wavenumber ($n \leq 4$) Rossby waves. This allows us to conclude that the tropospheric response of SSW and SPV events manifests itself mainly in that specific range of wavenumber/phase speed harmonics. Such a range of harmonics is similar to the one usually observed in the stratosphere, and would plausibly correspond to the tropospheric
impact of a stratospheric phenomenon such as SSW and SPV events. The enhancement of harmonics with high wavenumbers following SSW events is also consistent with previous research emphasizing the role of baroclinic eddies in inducing the large-scale circulation changes after such events.

**Outlook** This analysis was limited by the number of SSW and SPV events observed in the ERA5 data set. One possibility to increase the number of events is to utilize extended-range ensemble forecasts or seasonal hindcasts, hence larger data sets that
may yield more robust results through a higher number of SSW and SPV events. Such an approach has already been applied successfully to analyze the effect of SSWs on the Arctic Oscillation or North Atlantic Oscillation (Spaeth and Birner, 2022; Kolstad et al., 2022; Bett et al., 2023), and could yield further insights into the dynamics of weak and strong polar vortex events, for instance by partitioning them between split and displacement events (e.g., Fujiwara et al., 2022).

We stress here that the significant anomalies observed in the stratospheric spectra and co-spectra at harmonics that would be
"forbidden" by the Charney-Drazin criterion result from deviations of small absolute magnitude, but large relative magnitude with respect to the usual activity of such waves in the upper stratosphere. Given the associated positive heat flux contribution during SSW events, we speculate that such waves retain their baroclinic nature even if their amplitude exponentially decays as they ascend from the troposphere to the upper stratosphere. The possible contribution of such "unusual" small-scale waves to the dynamics of SSW and SPV events can be a matter of research, together with the effects of Rossby wave breaking across
different vertical levels or of mesovortices at the edge of the polar vortex (Waugh and Dritschel, 1999).



**Appendix A:  List of stratospheric extreme events**

| SSW events | | | SPV events | | |
|---|---|---|---|---|---|
| **start date** | difference [d] | peak date | start date | difference [d] | **peak date** |
| 1980-02-29 | 1 | 1980-03-01 | 1980-01-15 | 3 | 1980-01-18 |
| 1981-03-04 | 0 | 1981-03-04 | 1980-12-08 | **38** | 1981-01-15 |
| 1981-12-04 | 0 | 1981-12-04 | 1982-12-11 | **30** | 1983-01-10 |
| 1984-02-24 | 2 | 1984-02-26 | 1983-12-18 | **37** | 1984-01-24 |
| 1985-01-01 | 1 | 1985-01-02 | 1985-12-19 | 1 | 1985-12-20 |
| 1987-01-23 | 17 | 1987-02-09 | 1986-11-21 | 1 | 1986-11-22 |
| 1987-12-08 | 4 | 1987-12-12 | 1988-01-25 | 19 | 1988-02-13 |
| 1988-03-14 | 1 | 1988-03-15 | 1988-12-19 | **33** | 1989-01-21 |
| 1989-02-21 | 4 | 1989-02-25 | 1989-12-21 | 14 | 1990-01-04 |
| 1998-12-15 | 3 | 1998-12-18 | 1990-12-06 | 20 | 1990-12-26 |
| 1999-02-26 | 9 | 1999-03-07 | 1991-12-17 | 3 | 1991-12-20 |
| 2000-03-20 | 1 | 2000-03-21 | 1992-12-19 | 16 | 1993-01-04 |
| 2001-02-11 | 7 | 2001-02-18 | 1994-12-12 | 17 | 1994-12-29 |
| 2001-12-30 | 3 | 2002-01-02 | 1995-12-30 | **43** | 1996-02-11 |
| 2003-01-18 | 0 | 2003-01-18 | 1997-01-03 | 7 | 1997-01-10 |
| 2004-01-05 | 4 | 2004-01-09 | 1997-02-03 | 9 | 1997-02-12 |
| 2006-01-21 | 5 | 2006-01-26 | 1999-12-21 | 21 | 2000-01-11 |
| 2007-02-24 | 2 | 2007-02-26 | 2004-12-16 | **33** | 2005-01-18 |
| 2008-02-22 | 2 | 2008-02-24 | 2006-12-06 | 3 | 2006-12-09 |
| 2009-01-24 | 4 | 2009-01-28 | 2007-01-11 | 4 | 2007-01-15 |
| 2010-02-09 | 1 | 2010-02-10 | 2007-12-23 | 2 | 2007-12-25 |
| 2010-03-24 | 1 | 2010-03-25 | 2008-12-29 | 10 | 2009-01-08 |
| 2013-01-06 | 12 | 2013-01-18 | 2010-01-07 | 3 | 2010-01-10 |
| 2018-02-12 | 2 | 2018-02-14 | 2011-01-21 | **30** | 2011-02-20 |
| 2019-01-02 | 8 | 2019-01-10 | 2013-12-15 | 12 | 2013-12-27 |
| average | 3.76 | | average | 16.48 | |

**Table A1.** List of SSW and SPV events with start date, number of days between start date and peak date (difference), as well as peak date. Differences exceeding 27 days are shown in bold. The column of SSW start dates, as well as the column of SPV peak dates are used in the analysis. The last row depicts the average of the differences.



*Author contributions.*  This work originated from the Master Thesis by MS (Schutte, 2023), supervised by JR at Uppsala University between November 2022 and February 2023. MS analyzed the data and prepared the figures, while JR computed Rossby wave spectra and wrote the manuscript in collaboration with MS and DD. MS and JR share first authorship.

*Competing interests.*  At least one of the (co-)authors is a member of the editorial board of Weather and Climate Dynamics.

*Acknowledgements.*  We would like to thank Hilla Afargan-Gerstman for checking the consistency of SSW dates between ERA-Interim and ERA5, Heini Wernli and Gabriele Messori for useful discussions about the interpretation of the results. JR acknowledges funding from the Swiss National Science Foundation (SNSF) via grant PZ00P2_209135, and from the European Research Council (ERC) under the European

Union's Horizon 2020 research and innovation program (grant agreement no. 948309, CENÆ project, PI: Gabriele Messori). Support from the Swiss National Science Foundation through project PP00P2_198896 to D.D. is gratefully acknowledged.



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
