# Peer review of "Opposite spectral properties of Rossby waves during weak and strong stratospheric polar vortex events"

_EGUsphere, 2023_

## Author Comment (AC1)

**Response to comment by Anonymous Referee #1**

*This study provides a systematic characterization of Rossby wave activity during the 25 sudden stratospheric warming (SSW) and 31 strong polar vortex (SPV) events that occurred in the period 1979-2021 using space-time spectral analysis. The results reveal the Rossby wave behaviors during two situations of stratosphere-troposphere coupling. Overall, this study is very interesting and worth of publication on WCD. I only have some few concerns or comments.*

*General comments:*

*The significance of some results are not robust due to limited composite samples used here. Although the authors made efforts to comprehensively analyze these cases, I think they may use the ensemble runs derived from SNAPSI (Hitchcock et al., 2022) to support the key findings during SSW events, particularly for the 'rule-of-thumb' proposed by the authors. The authors may compare differences in the wave activities during control runs and nudged runs.*

Thanks for your general and specific comments and your time spent reading the paper. We agree that the use of ensemble runs could enhance the robustness of our results, as also pointed out in the outlook of our section 7. At the same time, we would like to stress that even the limited number of 31 SPV events and 25 SSW events identified in the ERA5 dataset for the period from 1979 to 2021 yielded in significantly different spectral patterns of Rossby waves between SSWs and SPVs in the stratosphere (cf. Fig. 3 and 5), and to some extent even in the troposphere (cf. Fig. 4 and 6). As also highlighted in several places in the manuscript (e.g., L.243 ff., L.263 ff., L331 ff.), the tropospheric response to stratospheric extremes appears to be more complex than the stratospheric one which is likely due to additional factors influencing tropospheric Rossby wave behavior (e.g., baroclinicity, adiabatic forcing) and not because of the limited number of events. Thus, the use of ensemble runs would probably not increase the robustness in the tropospheric response.

Additionally, we acknowledge that using ensemble runs derived from SNAPSI, could help attributing the effect of stratospheric extreme events on specific harmonics of Rossby wave spectra in the troposphere. However, this dataset includes only ensemble runs for 3 events, so some results might be very event-specific. The limitation of only 3 different events being represented in SNAPSI would therefore not really help with the sample size. Furthermore, our way of performing space-time spectral analysis requires time intervals of 60 days (see section 2.2). This could limit the data of spectra even further in context of ensemble runs derived from SNAPSI. Lastly, the analysis of the SNAPSI dataset would likely address a different set of research questions, as those proposed in the introduction of our manuscript (L60-62).

*In addition, as the authors said, using 48 m/s as a threshold value is arbitrary. I suggested replacing it with 1 standardize deviation of zonal wind climatology as the threshold to select SPV events.*

It is a good idea to use exceedance of 1 std of 60N zonal mean zonal wind as alternative definition of SPV events instead of the, on first sight, arbitrary value of 48 m/s. Our motivation for defining SPV events as times when zonal mean zonal winds exceed 48 m/s was motivated by making results comparable to existing literature about SPVs (Oehrlein et al., 2020; Scaife et al., 2016; Smith et al., 2018). These authors motivated the definition as an analogous way to the SSW definition of Charlton and Polvani (2007a), stating that the threshold of 48m/s is chosen as it is exceeded with the same frequency as the lower SSW threshold (Scaife et al., 2016). Since we aim to compare the impact of SSWs and SPVs respectively, it seemed like this analogous definition would be the fairest way of doing so.

Thus, even though 48 m/s might seem a rather arbitrary value, the motivation for choosing this specific threshold actually considers the variation, and could be seen as a modified way of standard deviation.

*Specific comments:*

*L6: concomitant -> simultaneous*

Thanks for pointing this out, we will change this.

*L44: complicated -> complex*

Good point, we will modify the following part in L44: "…analysis is complicated by…" → "… analysis becomes even more complex by…".

*L83: 'SSW events' should be 'SPV events'.*

Thanks for highlighting this. We actually meant SPV events here, as the whole paragraph aims to motivate the definition of SPV events. The definition of SSW events by Charlton and Polvani (2007a) is widely accepted and based on the physical threshold of a reversal of zonal winds. This connects well with the warming occurring simultaneously in the polar stratosphere due to the thermal wind balance.

*L163: What do you mean about this sentence 'periods around SSW and SPV events are not excluded'? Please clarify it.*

We mean that our resampling procedure considers the whole data set, including the times when SSWs and SPVs occurred. We just wanted to highlight this detail, since for certain research questions, one might exclude the dates of anomalies in the data set for assessing their statistical significance (see also comment of reviewer 2 on line 163).

*L181:The words 'rapidly' and 'rapid' are not appropriate. The description of 'transient wave' is enough to describe the faster wave than stationary wave.*

Thanks for suggesting a different phrasing here. We will adapt this in our manuscript.

*L191: 'We not here that also' -> 'Also note that'*

Good point. We will change that.

*L237: This may be related to stronger stratosphere-troposphere coupling over the North Atlantic Ocean. Please refer to Garfinkel et al (2013) and Zhang et al. (2022).*

Thanks for providing the additional references. We will add them in the manuscript.

*L258: Is the positive anomaly caused by the internal variability or the sample error of composite cases? Does it also appear in large-ensemble experiments? Please see my general comments.*

The composite for the 31 SPV events shows significantly positive anomalies for eastward propagating waves (Fig. 6). Our significance assessment using random resampling indicates that that the positive anomalies are larger than some internal variability. Regarding the positive anomalies observed already before the onset of SPVs (Fig. 6a), we understand this as an effect of coupling between upper troposphere and stratosphere and thus a speed-up of the polar vortex. Furthermore, our choice of onset day as date with maximum zonally averaged

winds for SPVs (see section 2.1) can explain signals observed before this date, since the threshold of 48 m/s is already exceeded prior to that date.

*L325:It is confused why the heat flux co-spectra with higher-wavenumber westward propagating waves for SSW events are so noticeable at 10hPa in the upper stratosphere at which there only occurs wave 1 and 2. Please give more explanations.*

Thanks for highlighting this point. These synoptic-scale features are likely due to the break-up and shredding of the polar vortex during the onset of SSWs. We will refer to this even earlier in context of Fig.3 (L224).

*L356:This sentence should correspond to Fig. 11a and d.*

Thanks for spotting this mistake. We will fix it.

*L366: Fig.11b should be Fig.11a*

Thanks for this suggestion. We actually refer here to the weak negative anomaly for eastward propagating waves in Fig. 11b contrasting the positive anomaly in Fig. 11e. So, Fig. 11b is indeed meant here.

*References:*

*Hitchcock, P., Butler, A.H., Charlton-Perez, A.J., Garfinkel, C.I., Stockdale, T.N., Anstey, J.A., Mitchell, D.M., Domeisen, D.I., Wu, T., Lu, Y., Mastrangelo, D., Malguzzi, P., Lin, H., Muncaster, R., Merryfield, B., Sigmond, M., Xiang, B., Jia, L., Hyun, Y., Oh, J., Specq, D., Simpson, I.R., Richter, J.H., Barton, C.A., Knight, J.R., Lim, E., & Hendon, H.H. (2022). Stratospheric Nudging And Predictable Surface Impacts (SNAPSI): a protocol for investigating the role of stratospheric polar vortex disturbances in subseasonal to seasonal forecasts. Geoscientific Model Development.*

*Garfinkel, C.I., Waugh, D.W., & Gerber, E.P. (2012). The Effect of Tropospheric Jet Latitude on Coupling between the Stratospheric Polar Vortex and the Troposphere. Journal of Climate, 26, 2077-2095.*

*Zhang, J., Zheng, H., Xu, M., Yin, Q., Zhao, S., Tian, W., & Yang, Z. (2022). Impacts of stratospheric polar vortex changes on wintertime precipitation over the northern hemisphere. Climate Dynamics, 1-17.*

References:

Charlton, A. J. and Polvani, L. M.: A New Look at Stratospheric Sudden Warmings. Part I: Climatology and Modeling Benchmarks, J. Climate, 20, 449–469, https://doi.org/10.1175/JCLI3996.1, 2007a.

Oehrlein, J., Chiodo, G., and Polvani, L. M.: The effect of interactive ozone chemistry on weak and strong stratospheric polar vortex events, Atmos. Chem. Phys., 20, 10 531–10 544, https://doi.org/10.5194/acp-20-10531-2020, 2020.

Scaife, A. A., Karpechko, A. Y., Baldwin, M. P., Brookshaw, A., Butler, A. H., Eade, R., Gordon, M., MacLachlan, C., Martin, N., Dunstone, N., and Smith, D.: Seasonal winter forecasts and the stratosphere, Atmos. Sci. Lett., 17, 51–56, https://doi.org/10.1002/asl.598, 2016.

Smith, K. L., Polvani, L. M., and Tremblay, L. B.: The Impact of Stratospheric Circulation Extremes on Minimum Arctic Sea Ice Extent, J. Climate, 31, 7169–7183, https://doi.org/10.1175/JCLI-D-17-0495.1, 2018.

---

## Author Comment (AC2)

**Response to comment by Anonymous Referee #2**

*Review for "Opposite spectral properties of Rossby waves during weak and strong stratospheric polar vortex events" by Schutte et al. WCD*

*This manuscript by Schutte et al. (2023) uses observations and a novel technique (which has not been applied before to SSWs) to examine the spectral properties of Rossby waves throughout the lifecycles of SSW events and strong polar vortex events. The technique picks up on large spectral changes throughout the lifecycle with SSWs being characterized by a reduction in the eastward phase speeds in the stratosphere before the event, along with an increase in westward phase speeds, before being associated with an overall reduction in all phase speeds sufficiently far after the onset. This is consistent with enhanced upward propagating waves before the onset (although their characterization of this as being due to stationary waves requires further explanation; see my point below), and planetary wave suppression after the onset throughout the stratosphere and troposphere. SPV events seemingly show opposite-signed results. Although the stratospheric response is very clear, the tropospheric response to such events is less clear using this method which I suspect is due to the small number of observed events and thus strong inter-event variability, particularly given the more chaotic nature of the tropospheric flow.*

*The paper is well-written and I found it novel and interesting to read. I have only minor comments and so that is my suggestion. I do hope the authors will use a longer dataset or model simulations to better test the method, though!*

Thank you so much for your feedback and the time spent reading the paper. We agree that using a longer data set could provide a great opportunity to test the method further and obtain more robust results through a higher number of SSW and SPV events. As mentioned in our conclusions, one could utilize extended-range ensemble forecasts or seasonal hindcasts in a comparable manner as Spaeth and Birner (2022); Kolstad et al. (2022); Bett et al. (2023). However, this would likely provide enough material for an additional paper, as one should also compare the results from the ensemble runs with the reanalysis data. Thus, we have decided to limit our analysis to the data covered by the ERA5 reanalysis data set between 1979 and 2021.

*Minor Comments:*

*Lines 46-49: This sentence is a bit too flippant in its summary of the role of planetary and synoptic waves in the surface impact.*

*A large consensus in the community is that something initially brings the stratospheric influence to the surface and maintains an exogenous forcing to the troposphere. The two main such processes are 1) related to the changes in the meridional circulation, initially by the wave-induced meridional circulation during the onset stage (the 'downward control' idea), but later by the radiatively-driven circulation due to the persistent lower-stratospheric anomalies (e.g., Thompson et al. 2006 whom you already cite, and White et al. 2022, JAS). On the other hand, planetary wave suppression throughout the entire column can also provide the continued exogenous forcing (e.g., Hitchcock and Simpson 2016, JAS; Hitchcock and Haynes 2016, GRL). However, to amplify the tropospheric response due to the above mechanisms, it is generally well-accepted that the synoptic wave feedback in the troposphere is necessary to yield the wind dipole associated with the negative NAO (the meridional circulation nor the planetary wave effect can yield the full dipole). Both Song and Robinson (2004) and the aforementioned White et al. (2022) show this downward response being caused by the meridional circulation, but with an additional effect by the planetary waves (though much stronger in*

*Song and Robinson and in Hitchcock and Simpson 2016), followed by a synoptic-wave-induced tropospheric amplification.*

*Although this is likely too detailed for your paper introduction, comparing the planetary and synoptic wave effects and determining which is the 'main modulator' of the surface response is not well-founded nor the correct to phrase it, as they both play important, but crucially, very different roles in maintaining the tropospheric response.*

Thanks for pointing out this weakness in the text. We have now improved this sentence to better reflect the existing literature. It now reads: "For instance, both planetary-scale and synoptic-scale waves are crucial for the tropospheric response to SSWs (Song and Robinson, 2004; White et al., 2022). While planetary wave propagation tends to be suppressed after SSWs (Hitchcock and Simpson 2016, JAS; Hitchcock and Haynes 2016, GRL), the tropospheric response can be amplified by synoptic wave feedbacks (Domeisen et al., 2013; White et al., 2020)."

*Lines 75-76: Did you also use the other parts of the SSW definition defined in Charlton and Polvani? I presume that because you used dates directly from the Butler et al. paper, that you did. But for self-containment maybe state the extra conditions for SSW identification mentioned in the Charlton and Polvani paper. For clarity, the definition makes sure that the winds return to westerly for a 20-day period post easterlies, and must also return to westerly before the end of winter (end of April, I think).*

Yes, these additional conditions are included in the selection of SSWs in the data set of Butler (2020). We will clarify this.

*Line 80: Presumably you excluded the 22nd Feb 1979 event because there are not 60 days in that calendar year before the event occurred? Maybe state it directly?*

Yes, that is the reason for not including that event. More precisely, we decided to go winter by winter and start from NDJFM 1979/80. Even though including spectra from February and March 1979 would be possible in theory, time lags of more than -20 days would have been not available. As we need to consider time lags of up to -30 days in other parts of our analysis (e.g., Fig.7 and 8), we decided to exclude the SSW on Feb. 22$^{nd}$ 1979. We will specify that in the manuscript.

*Line 84: Why is 48ms-1 chosen? As you state, it is arbitrary, but without reading the Oehrlein et al paper, I do not know why it has been chosen? How sensitive are your results to increases or decreases in that value? I would think that a better and less arbitrary criterion would be to use some kind of standard deviation definition (although perhaps that is what the cited paper did already do)*

It is a good idea to use exceedance of 1 std of 60N zonal mean zonal wind as alternative definition of SPV events instead of the rather arbitrary value of 48 m/s. Our motivation for defining SPV events as times when zonal mean zonal winds exceed 48 m/s was motivated by making results comparable to existing literature about SPVs (Oehrlein et al., 2020; Scaife et al., 2016; Smith et al., 2018). These authors motivated the definition as an analogous way to the SSW definition of Charlton and Polvani (2007a), stating that the threshold of 48m/s is chosen as it is exceeded with the same frequency as the lower SSW threshold (Scaife et al., 2016). Since we aim to compare the impact of SSWs and SPVs respectively, it seemed like this analogous definition would be the fairest way of doing so. Thus, even though 48 m/s might seem a rather arbitrary value, the motivation for choosing this specific threshold actually considers the variation, and could be seen as a modified way of standard deviation. We will clarify this choice in L84.

*Line 117: Why do you use an extra 10degree in the tropospheric average compared to the stratospheric? For consistency across levels, maybe the same should be used? It is fine to do what you have done, but maybe just clarify why you chose those different bands for the reader.*

Thanks for pointing out this unclarity. To include only Rossby waves connected to the mid-latitude jet stream in the troposphere we compute Rossby wave spectra between 35°N and 75°N at 250 hPa. Since the stratospheric polar vortex is located mostly between 45°N and 75°N, Rossby wave spectra are computed in the stratosphere for that latitude range. Thus, the different choice of latitudes can be justified by the different latitudinal range of the eddy-driven jet in the troposphere and the polar vortex in the stratosphere. We will add an explanation in this context in the paper.

*Lines 145-150: I think this quantity was plotted in figure 2 but it was not clear from the text or figure caption. Can you clarify? Perhaps define this hemispherically-averaged phase speed as a different variable or just make sure to refer to it properly in the figure (also figure 7, for instance).*

Thanks for highlighting this. We will refer to the definition of phase speed and ISP in the captions of the figures.

*Line 163: Is there a reason you did not exclude the SSW and SPV event days from the resampling procedure?*

We wanted to have a realistic data set of the atmosphere for our resampling procedure. This means, that also the extreme states (SSWs and SPVs) are included. As a result, also extreme states can be drawn for each random sample, which gives in our opinion a fair assessment of the statistical significance of the results.

*Line 213: You divide the anomaly for a particular SSW or SPV by that season's mean? Or by an overall climatological seasonal mean?*

We divide here by the overall climatological seasonal mean. Thanks for asking, we will clarify this in the text.

*Lines 222-224: To clarify, a stationary wave pattern is suggested because the anomalies are dominated heavily by westward-propagating wave anomalies with a reduction in the climatologically-dominated eastward-propagating anomalies, and so this would lead to a more stationary pattern? I see that you said the reduction of all wavenumbers simultaneously likely indicates a weakening of the background flow, but the 'stationary' part is not clear to me.*

A more stationary pattern was meant here in a way that Rossby waves are propagating at slower phase speeds than usual. Connected with the slow-down of the zonal mean wind, also the propagation of Rossby waves slows down, before Rossby wave activity is reduced across most wave numbers and phase speeds (Fig. 3d). We will clarify this in the manuscript.

*Figure 4 (and other relevant figures): Why did you choose 250hPa as the 'tropospheric' level? 250hPa is already in the lower stratosphere at sub-polar latitudes. I guess your latitudinal average of 35-75N will be dominated by the lower latitudes when weighted. But if you stick with 250hPa, can you address how sensitive the results in this and other figures are to surrounding levels?*

Thanks for pointing out the unclarity about the spectra at 250 hPa. This level was also chosen to analyze the impact of stratospheric extremes at 10 hPa on lower levels in the atmosphere, mainly the upper troposphere. We agree, that depending on the definition of the tropopause, the pressure level of 250 hPa can be indeed in the stratosphere. However, from a dynamical

perspective, the jet stream is a result of the gradient between polar air (with a lower tropopause) and sub-tropical air (with a higher tropopause). Even though the jet stream might be strongest directly below the tropopause, it is still present at levels above and below, and thus also Rossby waves. Furthermore, 250 hPa is also a way of compromise to investigate different effects at the same pressure level: In order to evaluate the spectra of meridional heat fluxes, Sjoberg and Birner (2012) actually advocated for 200hPa as a "fully stratospheric" pressure level, since $N^2$ is constant. On the other hand, 300 hPa is still half tropospheric and half stratospheric. Going to even lower levels, like 500 hPa, would result in analyzing the flow below the jet maximum. One can additionally argue that it is at the level of the tropopause that PV gradients and PV anomalies are the strongest and the most variable. Thus, 250 hPa seemed the most appropriate level for our analysis of tropospheric impact of stratospheric extremes at 10 hPa.

*Lines 232-239: This reduction in upward-propagating low-wavenumber waves strikes me as supporting the idea that planetary waves after an SSW onset become suppressed throughout the entire atmospheric column (not just the stratosphere) in agreement with Hitchcock and Haynes (2016: GRL) and Hitchcock and Simpson (2016; JAS).*

*Also, can you clarify why this reduction in the V'V' spectra provides evidence of a more persistent tropospheric flow post onset? I would have thought momentum fluxes would indicate this phenomenon. I understand V'V' to be more of an eddy kinetic energy-type quantity and given the reduction at low wavenumbers only, it indicates more of a planetary wave suppression in my eyes. I am happy to be proven wrong though!*

Thanks for pointing out that this supports the hypothesis of a reduction in planetary waves throughout the entire column. This is a very interesting point and could be investigated in a future study.

A persistent flow was meant here, similar to the "stationary pattern" in line 222-224, that the jet stream is located over the same region for a longer time than usual. Since planetary-scale waves are suppressed, only synoptic-scale disturbances can chance the position of the jet stream. These are, by definition, smaller and thus, the large-scale pattern is more persistent. This agrees with other research, e.g., a longer persistence of the negative phase of the NAO pattern (Charlton-Perez et al., 2018; Domeisen, 2019).

*Line 240: This looks to only be slightly significant at lags 10-20 days and is only sporadically positive at other lags.*

We agree, that this signal is relatively noisy. Furthermore, the lack of significance could also be due to compounding the 25 events, where the shift towards higher wave numbers might not happen equally for each event. Nevertheless, a shift towards higher wavenumbers is present for eastward propagating harmonics, especially 10 to 20d after the event (Fig. 4c). Another point that supports this tendency is the symmetry of the SPV response with respect to SSWs (cf. Fig. 4 and 6). This gives us further confidence in our interpretation.

*Lines 268-269: yes, but 250hPa is already in the lower portion of the vortex (see my point above).*

Let's see how similar the behavior a of lower level (e.g., 300 hPa) is compared to the chosen 250 hPa.

*Lines 273-279: This gradual reduction in the (significant) phase speed anomalies from the middle stratosphere to the lower stratosphere is reminiscent of the downward propagation of critical lines*

*from the upper to lower stratosphere post SSW, a la Matsuno (1971) and Hitchcock and Haynes (2016).*

Thanks for pointing out this connection. We will consider it in the revised version.

*Figure 9: Any ideas as to why the synoptic waves at k=4+ or so, also show a strengthening of westward phase speeds? To me, I wonder if it essentially represents the breaking up of the vortex and filamentation of potential vorticity streamers from the edge of the vortex as planetary waves break, leading to smaller-scale features. In any case, it would be good to mention the interesting, albeit confusing, synoptic-wave features so high up in the stratosphere.*

We agree with your impression. These synoptic-scale features are likely due to the break-up of the polar vortex. We discuss this later in lines 360-362. Thanks for highlighting that interpretation. We will stress it more in the manuscript in L224 and 256.

*Lines 322-323: Is this because the vortex has weakened (post SPV maximum) to an extent that it is more receptive to upward-propagating waves from below?*

This point is discussed in context with Fig. 11 in lines 373-379. We presume that this behavior is indeed connected to a deceleration of the polar vortex, even though the specific Rossby wave signature differs between pre-SSW and post-SPV.

*Grammatical Comments:*

*Lines 29-32: I would rewrite this opening sentence. It is a little confusing to read and disjointed (especially the part before the colon). The part after the colon I would just write as a new sentence.*

Thanks for highlighting this sentence. We will rewrite it.

*Lines 40-43: The opening sentence is again a bit disjointed. The NAM and AO you put in the first set of parentheses are zonally-averaged themselves, whereas you add about zonally-averaged quantities directly after, and using Hall et al. (2021) as an example. The Baldwin and Dunkerton 2001 and Thompson et al 2006 papers also use zonally-averaged quantities. Multiple sets of parentheses (in this case, 3!) can be confusing and break up the sentence too much. I suggest:*

*"The impact of SSW and SPV events on the tropospheric circulation is most often analyzed in terms of circulation indices of zonally averaged quantities (such as the Arctic Oscillation or the Northern Annular Mode, e.g., Baldwin and Dunkerton, 2001; Thompson et al., 2006), or in terms of changes in the frequency of weather regimes (e.g., Charlton-Perez et al., 2018; Domeisen et al., 2020c; Hall et al., 2023)."*

We agree. Your suggestion improves the readability of that sentence.

*Line 121: update end of line to: '...allow ONE to neglect...'*

Thanks, we will fix that.

*Line 214: change to '... a value twice as large. '*

Well spotted, we will change this.

*Line 359: 'reflect' --> 'lead to' or 'result in'*

Good point, we will adjust the wording.

*References:*

*Hitchcock, P., and Haynes, P. (2016), Stratospheric control of planetary waves, Geophys. Res. Lett., 43, 11,884–11,892, doi:10.1002/2016GL071372.*

*Hitchcock, P., and I. R. Simpson, 2016: Quantifying Eddy Feedbacks and Forcings in the Tropospheric Response to Stratospheric Sudden Warmings. J. Atmos. Sci., 73, 3641–3657, https://doi.org/10.1175/JAS-D-16-0056.1.*

*Matsuno, T. (1971), A dynamical model of the stratospheric sudden warming,J. Atmos. Sci.,28, 1479–1494.*

*White, I. P., C. I. Garfinkel, and P. Hitchcock, 2022: On the Tropospheric Response to Transient Stratospheric Momentum Torques. J. Atmos. Sci., 79, 2041–2058, https://doi.org/10.1175/JAS-D-21-0237.1.*

References:

Bett, P. E., Scaife, A. A., Hardiman, S. C., Thornton, H. E., Shen, X., Wang, L., and Pang, B.: Using large ensembles to quantify the impact of sudden stratospheric warmings and their precursors on the North Atlantic Oscillation, Weather Clim. Dynam., 4, 213–228, https://doi.org/10.5194/wcd-4-213-2023, 2023.

Butler, A.: Table of major mid-winter SSWs in reanalyses products, https://csl.noaa.gov/groups/csl8/sswcompendium/majorevents.html, 2020.

Charlton, A. J. and Polvani, L. M.: A New Look at Stratospheric Sudden Warmings. Part I: Climatology and Modeling Benchmarks, J. Climate, 20, 449–469, https://doi.org/10.1175/JCLI3996.1, 2007a.

Charlton-Perez, A. J., Ferranti, L., and Lee, R. W.: The influence of the stratospheric state on North Atlantic weather regimes, Q J Roy. Meteor. Soc., 144, 1140–1151, https://doi.org/10.1002/qj.3280, 2018.

Charlton-Perez, A. J., Ferranti, L., and Lee, R. W.: The influence of the stratospheric state on North Atlantic weather regimes, Q J Roy. Meteor. Soc., 144, 1140–1151, https://doi.org/10.1002/qj.3280, 2018.

Kolstad, E. W., Lee, S. H., Butler, A. H., Domeisen, D. I., and Wulff, C. O.: Diverse surface signatures of stratospheric polar vortex anomalies, J. Geophys. Res.-Atmos., 127, e2022JD037 422, 2022.

Oehrlein, J., Chiodo, G., and Polvani, L. M.: The effect of interactive ozone chemistry on weak and strong stratospheric polar vortex events, Atmos. Chem. Phys., 20, 10 531–10 544, https://doi.org/10.5194/acp-20-10531-2020, 2020.

Scaife, A. A., Karpechko, A. Y., Baldwin, M. P., Brookshaw, A., Butler, A. H., Eade, R., Gordon, M., MacLachlan, C., Martin, N., Dunstone, N., and Smith, D.: Seasonal winter forecasts and the stratosphere, Atmos. Sci. Lett., 17, 51–56, https://doi.org/10.1002/asl.598, 2016.

Smith, K. L., Polvani, L. M., and Tremblay, L. B.: The Impact of Stratospheric Circulation Extremes on Minimum Arctic Sea Ice Extent, J. Climate, 31, 7169–7183, https://doi.org/10.1175/JCLI-D-17-0495.1, 2018.

Spaeth, J. and Birner, T.: Stratospheric modulation of Arctic Oscillation extremes as represented by extended-range ensemble forecasts, Weather Clim. Dynam., 3, 883–903, https://doi.org/10.5194/wcd-3-883-2022, 2022.

White, I. P., C. I. Garfinkel, and P. Hitchcock: On the Tropospheric Response to Transient Stratospheric Momentum Torques. J. Atmos. Sci., 79, 2041–2058, https://doi.org/10.1175/JAS-D-21-0237.1, 2022.

---

## Author Response (AR1)

**Opposite spectral properties of Rossby waves during weak and strong stratospheric polar vortex events**

Michael Schutte, Jacopo Riboldi, Daniela Domeisen

Uppsala, December 15, 2023

**Submitted to Weather and Climate Dynamics**

**Authors' response**

We would like to thank the two anonymous reviewers for their helpful comments on the manuscript. The most important changes made in the revision of the manuscript are listed below:

1. The paragraph in the Introduction regarding coupling between the stratosphere and the troposphere has been re-written following suggestions by the reviewers.

2. The adopted SPV definition has been further clarified. Additionally, we checked the robustness of the results to an alternative choice of the threshold (mean+1 standard deviation) for SPV events (43m/s; see answer to first reviewer's second general comment).

3. The role of small-scale disturbances during the onset of SSWs, and lack thereof during onset of SPVs, was highlighted.

4. The sensitivity of the computed spectra to the choice of the vertical level was tested using an alternative level located mostly in the troposphere, at 300 hPa. As results are very similar to the original choice of $250\,$hPa, we decided to keep the latter as an upper-tropospheric level capable of capturing Rossby wave dynamics at the level of the tropopause.

In the following, the referee's comments are in *black and italic* and our answers are below each of them in **bold blue**. We further indicate adjustments made to the manuscript in parentheses below our answers where applicable. All line numbers refer to the revised manuscript.

**1 Response to comment by Anonymous Referee #1**

*This study provides a systematic characterization of Rossby wave activity during the 25 sudden stratospheric warming (SSW) and 31 strong polar vortex (SPV) events that occurred in the period 1979-2021 using space-time spectral analysis. The results reveal the Rossby wave behaviors during two situations of stratosphere-troposphere coupling. Overall, this study is very interesting and worth of publication on WCD. I only have some few concerns or comments.*

*General comments:*

*The significance of some results are not robust due to limited composite samples used here. Although the authors made efforts to comprehensively analyze these cases, I think they may use the ensemble runs derived from SNAPSI (Hitchcock et al., 2022) to support the key findings during SSW events, particularly for the 'rule-of-thumb' proposed by the authors. The authors may compare differences in the wave activities during control runs and nudged runs.*

We agree that the use of ensemble runs could enhance the robustness of our results, as also pointed out in the outlook of our section 7. At the same time, we would like to stress that even the limited number of 31 SPV events and 25 SSW events identified in the ERA5 dataset for the period from 1979 to 2021 yielded significantly different spectral patterns of Rossby waves between SSWs and SPVs in the stratosphere (cf. Fig. 3 and 5), and to some extent even in the troposphere (cf. Fig. 4 and 6). As also highlighted in several places in the manuscript (e.g., L.253 ff., L.278 ff., L348 ff.), the tropospheric response to stratospheric extremes appears to be more complex than the stratospheric one. This is likely due to additional factors influencing tropospheric Rossby wave behavior (e.g., baroclinicity, non-adiabatic forcing) rather than the limited number of events (e.g., Chan et Plumb, 2009; Afargan-Gerstman et al., 2022). Even current ensembles do not represent the full width of options (e.g., Kolstad et al., 2022). Thus, we believe that the use of ensemble runs would not necessarily increase the robustness in the tropospheric response.

Even though we agree that using ensemble runs derived from SNAPSI could potentially help attributing the effect of stratospheric extreme events on specific harmonics of Rossby wave spectra in the troposphere, we also note that the SNAPSI dataset includes only ensemble runs for 3 events: thus, the results will be very event-specific and not representative of the diversity of events. The analysis of the 3 different events represented in SNAPSI would therefore not help with the sample size. Furthermore, our way of performing space-time spectral analysis requires time intervals of 60 days (see section 2.2). This could limit the time period of usable data of spectra even further in the context of ensemble runs derived from SNAPSI. In conclusion, even though we appreciate the suggestion of the reviewer, the analysis of the SNAPSI dataset would likely address a different set of research questions than those proposed in the introduction of our manuscript (lines 60-62).

[Figure]

**Figure 1.** Relative anomalies of spectral power $S_{\overline{V'V'}}$ compared to the mean spectrum at 250 hPa, averaged over 10-day time intervals, around SPV events (shaded). Subplot (a) shows spectra for 10 days prior to the event to the event start, subplot (b) shows spectra between the event start and 10 days after the event start, (c) for 10 to 20 days, and (d) for 20 to 30 days after the event. Wavenumber-phase speed pairs marked with an × exceed the 0.5th or 99.5th percentile of the re-sampled distribution. Black contour lines show the NDJFM climatology ranging from $0.1\,\mathrm{m^2\,s^{-2}\,\Delta c^{-1}}$ to $0.8\,\mathrm{m^2\,s^{-2}\,\Delta c^{-1}}$ in steps of $0.1\,\mathrm{m^2\,s^{-2}\,\Delta c^{-1}}$, as Fig. 6 in the manuscript. The four left sub-panels show the original Figure applying $48\,\mathrm{m\,s^{-1}}$ as threshold, the four to the right with the alternative definition of 1 std ($43\,\mathrm{m\,s^{-1}}$).

*In addition, as the authors said, using 48 m/s as a threshold value is arbitrary. I suggested replacing it with 1 standardize deviation of zonal wind climatology as the threshold to select SPV events.*

We agree with the reviewer that it is a good idea to use exceedance of 1 std of 60N zonal mean zonal wind as alternative definition of SPV events instead of the, on first sight, arbitrary value of 48 m/s. Our motivation for defining SPV events as times when zonal mean zonal winds exceed 48 m/s was motivated by making results comparable to the existing literature about SPVs (Oehrlein et al., 2020; Scaife et al., 2016; Smith et al., 2018). These authors motivated the threshold-based SPV definition as analogous to the SSW definition of Charlton and Polvani (2007a), stating that the threshold of 48 m/s is chosen as it is exceeded with the same frequency as the lower SSW threshold (Scaife et al., 2016). Since we aim to compare the impact of SSWs and SPVs respectively, it seemed like this analogous definition would be the fairest way of doing so. This analogy with SSWs is now specified in the manuscript at lines 86-88, where the following sentences have been added: "Scaife et al. (2016) state that the threshold of $48\,\mathrm{m\,s^{-1}}$ is chosen as it is exceeded with the same frequency as the lower SSW threshold. Thus, the SPV definition can be seen as analogous to the threshold-based SSW definition of Charlton and Polvani (2007) and aims for making results of SSWs and SPVs comparable with each other."

Nevertheless, we made a comparison between the two definitions, using the exceedance of the 48 m/s threshold or alternatively of 1 std with respect to the NDJFM mean (new threshold: $43.02\,\mathrm{m\,s^{-1}}$). The latter choice results in the definition of 34 SPV events, three more than with the 48m/s threshold.

[Figure]

**Figure 2.** Time series of daily averages around SSW (a) and SPV (b) events for phase speed anomaly with respect to the NDJFM seasonal cycle at each level. Points marked with an × lie outside of the 99% confidence interval with respect to the seasonal cycle, as Fig. 6 in the manuscript. The left sub-panel shows the original figure applying 48 $\mathrm{m\,s^{-1}}$ as threshold, the right sub-panel with the alternative definition of 1 std (43 $\mathrm{m\,s^{-1}}$).

The comparison shows only minor differences in the results (Fig. 1 and 2). This additional sensitivity check confirms that the initial choice of 48 m/s as threshold is appropriate.

*Specific comments:*

*L6: concomitant -> simultaneous*

Thank you for pointing this out, we modified it as suggested (line 6).

*L44: complicated -> complex*

That's a good point, changed as suggested. (line 44: "Such an analysis is complicated by..." → "Such an analysis becomes even more complex by...")

*L83: 'SSW events' should be 'SPV events'.*

We actually meant SSW in the beginning of that paragraph. The formulation "As for SSWs," is employed because we would like to draw a parallel between the SPV and the SSW event definition, in the sense that SPV events are also defined from the 60°N zonally averaged zonal wind, and with respect to the exceedence of a threshold. The definition of SSW events by Charlton and Polvani (2007a) is widely accepted and based on the physical threshold of a reversal of zonal winds.

*L163: What do you mean about this sentence 'periods around SSW and SPV events are not excluded'? Please clarify it.*

We mean that our re-sampling procedure considers the whole data set, including the times when SSWs and SPVs occurred. We just wanted to highlight this detail, since for certain research questions, one might exclude the dates

of anomalies in the data set for assessing their statistical significance (see also comment of reviewer 2 on line 163). We anyway modified the sentence pointed out, as it can result misleading, and now explicitly state that "SSW and SPV events are not excluded from the re-sampling" (line 167).

*L181:The words 'rapidly' and 'rapid' are not appropriate. The description of 'transient wave' is enough to describe the faster wave than stationary wave.*

Thanks for suggesting a different phrasing here. We removed the word "rapid" from "rapid transients" (lines 188, 193) and modified the formulation through the manuscript at lines 186,304.

*L191: 'We not here that also' -> 'Also note that'*

Thanks once more for the suggestion, that we followed (lines 195-196: "We not here that also" → "Also note that").

*L237: This may be related to stronger stratosphere-troposphere coupling over the North Atlantic Ocean. Please refer to Garfinkel et al (2013) and Zhang et al. (2022).*

Thank you for providing those additional references, we added them to the manuscript together with your expressed idea of a stronger stratosphere-troposphere coupling over the North Atlantic Ocean (lines 247-248: "This could be related to the generally stronger stratosphere-troposphere coupling observed over that region (Garfinkel et al., 2013 and Zhang et al., 2022).

*L258: Is the positive anomaly caused by the internal variability or the sample error of composite cases? Does it also appear in large-ensemble experiments? Please see my general comments.*

The composite for the 31 SPV events shows significantly positive anomalies for eastward propagating waves (Fig. 6). Our significance assessment using random re-sampling indicates significantly positive anomalies of the composite more than 0.3 standard deviations above the climatological mean. Since the re-sampling procedure, described in section 2.4, uses the data of all NDJFM extended winters between 1979 and 2021 we conclude that this is even significant with respect to internal variability between the different years. Regarding the positive anomalies observed already before the onset of SPVs (Fig. 6a), we understand this as an effect of coupling between the upper troposphere and the stratosphere and thus a speed-up of the polar vortex. Furthermore, our choice of the day with maximum zonally averaged winds as SPV event date (see section 2.1) means that SPV events defined by the threshold of 48 m/s will happen to that date, and hence our composites will contain SPV-like signals that are observed before the maximum wind speed is reached. We added the following sentence to the revised manuscript to make the reader aware of this point (lines 271-272): "Positive anomalies for eastward-propagating waves occur in the upper troposphere already in the days prior to the SPV event peak. This early signal is partly explained by the adopted definition of SPV

**event, defined in this study with respect to the day of maximum 10 hPa zonal wind at 60N".**

*L325:It is confused why the heat flux co-spectra with higher-wavenumber westward propagating waves for SSW events are so noticeable at 10hPa in the upper stratosphere at which there only occurs wave 1 and 2. Please give more explanations.*

**Thanks for highlighting this point. These synoptic-scale features are likely due to the break-up and filamentation of the polar vortex during the onset of SSWs. These features are, on the other hand, suppressed during SPV events because of the compact, intense polar vortex. We added to the text at lines 227-228: "The significant anomalies in spectral energy density visible for high-wavenumber, retrograding waves indicate small-scale disturbances in the polar vortex, which are likely the result of vortex filamentation during SSW events."**

*L356:This sentence should correspond to Fig. 11a and d.*

**Thank you for spotting this mistake. (line 373: (Figs. 11a,b) → (Figs 11a,d))**

*L366: Fig.11b should be Fig.11a*

**We actually refer here to the weak negative anomaly for eastward propagating waves in Fig. 11b contrasting the positive anomaly in Fig. 11e. We clarified the comparison in the manuscript in line 383.**

*References:*

*Hitchcock, P., Butler, A.H., Charlton-Perez, A.J., Garfinkel, C.I., Stockdale, T.N., Anstey, J.A., Mitchell, D.M., Domeisen, D.I., Wu, T., Lu, Y., Mastrangelo, D., Malguzzi, P., Lin, H., Muncaster, R., Merryfield, B., Sigmond, M., Xiang, B., Jia, L., Hyun, Y., Oh, J., Specq, D., Simpson, I.R., Richter, J.H., Barton, C.A., Knight, J.R., Lim, E., & Hendon, H.H. (2022). Stratospheric Nudging And Predictable Surface Impacts (SNAPSI): a protocol for investigating the role of stratospheric polar vortex disturbances in subseasonal to seasonal forecasts. Geoscientific Model Development.*
*Garfinkel, C.I., Waugh, D.W., & Gerber, E.P. (2012). The Effect of Tropospheric Jet Latitude on Coupling between the Stratospheric Polar Vortex and the Troposphere. Journal of Climate, 26, 2077-2095.*
*Zhang, J., Zheng, H., Xu, M., Yin, Q., Zhao, S., Tian, W., & Yang, Z. (2022). Impacts of stratospheric polar vortex changes on wintertime precipitation over the northern hemisphere. Climate Dynamics, 1-17.*

**References:**

Afargan-Gerstman, H., B. Jiménez-Esteve, and D. I. V. Domeisen, 2022: On the Relative Importance of Stratospheric and Tropospheric Drivers for the North Atlantic Jet Response to Sudden Stratospheric Warming Events. J. Climate, 35, 6453–6467, https://doi.org/10.1175/JCLI-D-21-0680.1.

Chan, C. J., and R. A. Plumb, 2009: The Response to Stratospheric Forcing and Its Dependence on the State of the Troposphere. J. Atmos. Sci., 66, 2107–2115, https://doi.org/10.1175/2009JAS2937.1.

Charlton, A. J. and Polvani, L. M.: A New Look at Stratospheric Sudden Warmings. Part I: Climatology and Modeling Benchmarks, J. Climate, 20, 449–469, https://doi.org/10.1175/JCLI3996.1, 2007a.

Kolstad, E. W., Lee, S. H., Butler, A. H., Domeisen, D. I. V., & Wulff, C. O., 2022: Diverse surface signatures of stratospheric polar vortex anomalies. Journal of Geophysical Research: Atmospheres, 127, e2022JD037422. https://doi.org/10.1029/2022JD037422

Oehrlein, J., Chiodo, G., and Polvani, L. M.: The effect of interactive ozone chemistry on weak and strong stratospheric polar vortex events, Atmos. Chem. Phys., 20, 10 531–10 544, https://doi.org/10.5194/acp-20-10531-2020, 2020.

Scaife, A. A., Karpechko, A. Y., Baldwin, M. P., Brookshaw, A., Butler, A. H., Eade, R., Gordon, M., MacLachlan, C., Martin, N., Dunstone, N., and Smith, D.: Seasonal winter forecasts and the stratosphere, Atmos. Sci. Lett., 17, 51–56, https://doi.org/10.1002/asl.598, 2016.

Smith, K. L., Polvani, L. M., and Tremblay, L. B.: The Impact of Stratospheric Circulation Extremes on Minimum Arctic Sea Ice Extent, J. Climate, 31, 7169–7183, https://doi.org/10.1175/JCLI-D-17-0495.1, 2018.

**2 Response to comment by Anonymous Referee #2**

*Review for "Opposite spectral properties of Rossby waves during weak and strong strato-spheric polar vortex events" by Schutte et al. WCD*

*This manuscript by Schutte et al. (2023) uses observations and a novel technique (which has not been applied before to SSWs) to examine the spectral properties of Rossby waves throughout the lifecycles of SSW events and strong polar vortex events. The technique picks up on large spectral changes throughout the lifecycle with SSWs being characterized by a reduction in the eastward phase speeds in the stratosphere before the event, along with an increase in westward phase speeds, before being associated with an overall reduction in all phase speeds sufficiently far after the onset. This is consistent with enhanced upward propagating waves before the onset (although their characterization of this as being due to stationary waves requires further explanation; see my point below), and planetary wave suppression after the onset throughout the stratosphere and troposphere. SPV events seemingly show opposite-signed results. Although the stratospheric response is very clear, the tropospheric response to such events is less clear using this method which I suspect is due to the small number of observed events and thus strong inter-event variability, particularly given the more chaotic nature of the tropospheric flow.*

*The paper is well-written and I found it novel and interesting to read. I have only minor comments and so that is my suggestion. I do hope the authors will use a longer dataset or model simulations to better test the method, though!*

**Thank you so much for your feedback. We agree that using a longer data set could provide a great opportunity to test the method further and obtain more robust results through a higher number of SSW and SPV events. As mentioned in our conclusions, one could utilize extended-range ensemble forecasts or seasonal hindcasts in a comparable manner as Spaeth and Birner (2022); Kolstad et al. (2022); Bett et al. (2023). However, this would likely provide enough material for an additional paper, as one should also compare the results from the ensemble runs with the reanalysis data. Thus, we have decided to limit our analysis to the data covered by the ERA5 reanalysis data set between 1979 and 2021.**

*Minor Comments:*

*Lines 46-49: This sentence is a bit too flippant in its summary of the role of planetary and synoptic waves in the surface impact.*

*A large consensus in the community is that something initially brings the stratospheric influence to the surface and maintains an exogenous forcing to the troposphere. The two main such processes are 1) related to the changes in the meridional circulation, initially by the wave-induced meridional circulation during the onset stage (the 'downward control' idea), but later by the radiatively-driven circulation due to the persistent lower-stratospheric anomalies (e.g., Thompson et al. 2006 whom you already cite, and White et al. 2022, JAS). On the other hand, planetary wave suppression throughout the entire column can also provide the continued exogenous forcing (e.g., Hitchcock and Simpson 2016,*

*JAS; Hitchcock and Haynes 2016, GRL). However, to amplify the tropospheric response due to the above mechanisms, it is generally well-accepted that the synoptic wave feedback in the troposphere is necessary to yield the wind dipole associated with the negative NAO (the meridional circulation nor the planetary wave effect can yield the full dipole). Both Song and Robinson (2004) and the aforementioned White et al. (2022) show this downward response being caused by the meridional circulation, but with an additional effect by the planetary waves (though much stronger in Song and Robinson and in Hitchcock and Simpson 2016), followed by a synoptic-wave-induced tropospheric amplification.*

*Although this is likely too detailed for your paper introduction, comparing the planetary and synoptic wave effects and determining which is the 'main modulator' of the surface response is not well-founded nor the correct to phrase it, as they both play important, but crucially, very different roles in maintaining the tropospheric response.*

**Thanks for pointing out this weakness in the text. We have now improved this sentence to better reflect the existing literature. It now reads in lines 46-49: "For instance, both planetary-scale and synoptic-scale waves are crucial for the tropospheric response to SSWs (Song and Robinson, 2004; White et al., 2022). While planetary wave propagation tends to be suppressed after SSWs (Hitchcock and Simpson 2016, JAS; Hitchcock and Haynes 2016, GRL), the tropospheric response can be amplified by synoptic wave feedbacks (Domeisen et al., 2013; White et al., 2020)."**

*Lines 75-76: Did you also use the other parts of the SSW definition defined in Charlton and Polvani? I presume that because you used dates directly from the Butler et al. paper, that you did. But for self-containment maybe state the extra conditions for SSW identification mentioned in the Charlton and Polvani paper. For clarity, the definition makes sure that the winds return to westerly for a 20-day period post easterlies, and must also return to westerly before the end of winter (end of April, I think).*

**Yes, these additional conditions are included in the selection of SSWs in the data set of Butler (2020). The condition that SPV events have to be separated by at least 20 days mimics this criterion, too. The following sentence has been added (lines 76-77): "Furthermore, following Charlton and Polvani (2007), they must return to westerly for more than 20 days after the SSW and before the end of the winter."**

*Line 80: Presumably you excluded the 22nd Feb 1979 event because there are not 60 days in that calendar year before the event occurred? Maybe state it directly?*

**Yes, that is the reason for not including that event. More precisely, we decided to go winter by winter and start from NDJFM 1979/80. Even though including spectra from February and March 1979 would be possible in theory, time lags of more than -20 days would have been not available. As we need to consider time lags of up to -30 days in other parts of our analysis (e.g., Fig.7 and 8), we decided to exclude the SSW on Feb. 22nd 1979. We will specify that in the manuscript. The sentence at lines 80-82 has been modified as follows: "Excluding the SSW event in February 1979, as Rossby wave spectra were only**

computed for full winters, starting in November 1979 onwards (see following section), the remaining 25 events constitute our set of SSWs in the following analysis."

*Line 84: Why is 48ms-1 chosen? As you state, it is arbitrary, but without reading the Oehrlein et al paper, I do not know why it has been chosen? How sensitive are your results to increases or decreases in that value? I would think that a better and less arbitrary criterion would be to use some kind of standard deviation definition (although perhaps that is what the cited paper did already do)*

**Thanks for pointing out this unclarity. Our motivation for defining SPV events as times when zonal mean zonal winds exceed 48 m/s was motivated by making results comparable to existing literature about SPVs (Oehrlein et al., 2020; Scaife et al., 2016; Smith et al., 2018). These authors motivated the definition as an analogous way to the SSW definition of Charlton and Polvani (2007a), stating that the threshold of 48 m/s is chosen as it is exceeded with the same frequency as the lower SSW threshold (Scaife et al., 2016). Since we aim to compare the impact of SSWs and SPVs respectively, it seemed like such a definition would be the fairest way of doing so. We have added to the text a sentence explaining the reason behind this threshold choice (line 86-88): "Scaife et al. (2016) state that the threshold of $48\,\mathrm{m\,s^{-1}}$ is chosen as it is exceeded with the same frequency as the lower SSW threshold. Thus, the SPV definition can be seen as analogous to the threshold-based SSW definition of Charlton and Polvani (2007) and aims for making results of SSWs and SPVs comparable with each other."**

**Following the recommendation of reviewer 1, we also tested the sensitivity of the results to a different threshold choice: the mean value of 60N, 10 hPa wind summed by 1 standard deviation. Results do not qualitatively change. Please see our answer to the first reviewer's second general comment for more details.**

*Line 117: Why do you use an extra 10degree in the tropospheric average compared to the stratospheric? For consistency across levels, maybe the same should be used? It is fine to do what you have done, but maybe just clarify why you chose those different bands for the reader.*

**Thanks for pointing out this unclear point. Rossby waves connected to the mid-latitude jet stream in the upper-troposphere have quite a broad latitudinal extension, as testified by the different polar and the subtropical waveguides. Thus, we compute Rossby wave spectra between 35°N and 75°N at 250 hPa to take into account this variability as much as possible, without leaving the midlatitudes. Since the stratospheric polar vortex is located mostly between 45°N and 75°N, and its strength is determined from the 60°N zonally averaged zonal wind, Rossby wave spectra are computed in the stratosphere for that latitude range. Thus, the different choice of latitudes can be justified by the different latitudinal range of the eddy-driven jet in the troposphere and the polar vortex in the stratosphere. This explanation has been added to the**

revised manuscript: (lines 121-122: "These are 35°N-75°N for the troposphere and 45°N-75°N in the stratosphere due to a different latitudinal extent of the tropospheric mid-latitude jet stream and the stratospheric polar vortex.")

*Lines 145-150: I think this quantity was plotted in figure 2 but it was not clear from the text or figure caption. Can you clarify? Perhaps define this hemispherically-averaged phase speed as a different variable or just make sure to refer to it properly in the figure (also figure 7, for instance).*

Thanks for highlighting this. We now refer to the definition of phase speed and ISP in the captions of figures 2, 7 and 8.

*Line 163: Is there a reason you did not exclude the SSW and SPV event days from the resampling procedure?*

We wanted to have a realistic data set of the atmosphere for our re-sampling procedure. This means, that also the extreme states (SSWs and SPVs) are included. As a result, also extreme states can be drawn for each random sample, which gives in our opinion a fair assessment of the statistical significance of the results.

*Line 213: You divide the anomaly for a particular SSW or SPV by that season's mean? Or by an overall climatological seasonal mean?*

We divide here by the overall climatological seasonal mean. Thanks for asking, we will clarify this in the text. The sentence reads now (lines 217-219): "For spectra, relative anomalies are computed by dividing the value of the anomaly by the NDJFM (November to March) seasonal mean averaged between all years from 1979 to 2021 for each wavenumber-phase speed harmonic."

*Lines 222-224: To clarify, a stationary wave pattern is suggested because the anomalies are dominated heavily by westward-propagating wave anomalies with a reduction in the climatologically-dominated eastward-propagating anomalies, and so this would lead to a more stationary pattern? I see that you said the reduction of all wavenumbers simultaneously likely indicates a weakening of the background flow, but the 'stationary' part is not clear to me.*

A "more stationary pattern" was meant here in a way that Rossby waves are propagating at slower phase speeds than usual. The deceleration of the zonal mean wind implies that the propagation of Rossby waves slows down, too, before Rossby wave activity is reduced across most wave numbers and phase speeds (Fig. 3d). (lines 228-230: "The stratospheric response to SSW events can be interpreted as an overall shift in Rossby wave activity towards a more stationary pattern, in the sense of a weakening of the climatological eastward propagation:")

*Figure 4 (and other relevant figures): Why did you choose 250hPa as the 'tropospheric' level? 250hPa is already in the lower stratosphere at sub-polar latitudes. I guess your lat-*

[Figure]

**Figure 3.** Relative anomalies of spectral power $S_{\overline{V'V'}}$ compared to the mean spectrum at 250 hPa, averaged over 10-day time intervals, around SSW events (shaded). Subplot (a) shows spectra for 10 days prior to the event to the event start, subplot (b) shows spectra between the event start and 10 days after the event start, (c) for 10 to 20 days, and (d) for 20 to 30 days after the event. Wavenumber-phase speed pairs marked with an × exceed the 0.5th or 99.5th percentile of the re-sampled distribution. Black contour lines show the NDJFM climatology ranging from $0.1\,\mathrm{m^2\,s^{-2}\,\Delta c^{-1}}$ to $0.8\,\mathrm{m^2\,s^{-2}\,\Delta c^{-1}}$ in steps of $0.1\,\mathrm{m^2\,s^{-2}\,\Delta c^{-1}}$, as Fig. 4 in the manuscript. The four left sub-panels show the original Figure at 250 hPa, the four to the right with the alternative level of 300 hPa.

*itudinal average of 35-75N will be dominated by the lower latitudes when weighted. But if you stick with 250hPa, can you address how sensitive the results in this and other figures are to surrounding levels?*

Thanks for raising this point. The 250 hPa level was chosen to analyze the impact of stratospheric extremes at 10 hPa on lower levels in the atmosphere, mainly the upper troposphere. We agree, that depending on the definition of the tropopause, the pressure level of 250 hPa can be indeed in the stratosphere. However, from a dynamical perspective, the jet stream is a result of the gradient between polar air (with a lower tropopause) and sub-tropical air (with a higher tropopause). Even though the jet stream might be strongest directly below the tropopause, it is still present at levels above and below, and thus also Rossby waves. Furthermore, 250 hPa is also a way of compromise to investigate different effects at the same pressure level: In order to evaluate the spectra of meridional heat fluxes, Sjoberg and Birner (2012) actually advocated for 200hPa as a "fully stratospheric" pressure level, since $N^2$ is constant. On the other hand, 300 hPa is still half tropospheric and half stratospheric. Going to even lower levels, like 500 hPa, would result in analyzing the flow below the jet maximum. One can additionally argue that it is at the level of the tropopause that PV gradients and PV anomalies are the strongest and the most variable. Thus, 250 hPa seemed the most appropriate level for our analysis of tropospheric impact of stratospheric extremes at 10 hPa.

We performed a comparison of Rossby wave spectra computed at 250 hPa and 300 hPa, which shows only minimal differences in the results (Fig. 3 for the stratosphere, as well as Fig. 4 for the troposphere). Thus, we will keep the

[Figure]

**Figure 4.** Same as Fig. 3, but for SPV events.

initially chosen 250 hPa level.

*Lines 232-239: This reduction in upward-propagating low-wavenumber waves strikes me as supporting the idea that planetary waves after an SSW onset become suppressed throughout the entire atmospheric column (not just the stratosphere) in agreement with Hitchcock and Haynes (2016: GRL) and Hitchcock and Simpson (2016; JAS).*

**Thanks for pointing out that this supports the hypothesis of a reduction in planetary waves throughout the entire column. This is a very interesting point and could be investigated in a future study. We added at lines 248-250 in the manuscript the following sentence: "Furthermore, the reduction of planetary wave activity at 250 hPa after SSWs supports the idea that planetary waves are suppressed throughout the entire atmospheric column after a SSW (Hitchcock and Haynes, 2016; Hitchcock and Simpson, 2016)."**

*Also, can you clarify why this reduction in the V'V' spectra provides evidence of a more persistent tropospheric flow post onset? I would have thought momentum fluxes would indicate this phenomenon. I understand V'V' to be more of an eddy kinetic energy-type quantity and given the reduction at low wavenumbers only, it indicates more of a planetary wave suppression in my eyes. I am happy to be proven wrong though!*

**We agree with you, that the reduction in V'V' spectra at positive phase speeds and low wavenumbers indicates asuppression of eastward propagating planetary waves. A "more persistent flow" was meant here, similar to the "more stationary pattern" in lines 229-230 (see answer to earlier comment), that Rossby waves are propagating at slower phase speeds than usual.**

*Line 240: This looks to only be slightly significant at lags 10-20 days and is only sporadically positive at other lags.*

**We agree that this signal is relatively noisy. Furthermore, the lack of significance could also be due to compounding the 25 events, where the shift towards**

higher wave numbers might not happen equally for each event. Nevertheless, a shift towards higher wavenumbers is present for eastward propagating harmonics, especially 10 to 20d after the event (Fig. 4c). Another point that supports this tendency is the symmetry of the SPV response with respect to SSWs (cf. Fig. 4 and 6). This gives us further confidence in our interpretation.

*Lines 268-269: yes, but 250hPa is already in the lower portion of the vortex (see my point above).*

Since there is no big difference between the spectra at 250 hPa and 300 hPa (see answer to comment on Fig. 4), this statement should be still vaild.

*Lines 273-279: This gradual reduction in the (significant) phase speed anomalies from the middle stratosphere to the lower stratosphere is reminiscent of the downward propagation of critical lines from the upper to lower stratosphere post SSW, a la Matsuno (1971) and Hitchcock and Haynes (2016).*

Thanks for pointing out this connection. We consider it in the revised version in lines 294-296. (Added sentence: "This agrees with other research pointing out the downward propagation of critical lines from the upper to lower stratosphere after SSW events (e.g., Matsuno, 1971; Hitchcock and Haynes, 2016)."

*Figure 9: Any ideas as to why the synoptic waves at k=4+ or so, also show a strengthening of westward phase speeds? To me, I wonder if it essentially represents the breaking up of the vortex and filamentation of potential vorticity streamers from the edge of the vortex as planetary waves break, leading to smaller-scale features. In any case, it would be good to mention the interesting, albeit confusing, synoptic-wave features so high up in the stratosphere.*

Thank you very much for providing this idea, we actually agree with your interpretation. These synoptic-scale features are likely due to the break up of the polar vortex. We discuss this now at lines 227-228 and 263-265. (Add in lines 227-228: " The significant anomalies in spectral energy density visible for high-wavenumber, retrograding waves indicate small-scale disturbances in the polar vortex, which are likely the result of vortex filamentation during SSW events."; Add in lines 263-265: "In contrast to SSW events, where positive small-scale anomalies likely result from filamentation, we interpret the negative anomalies at higher wave numbers during the onset of SPV as a sign of a more laminar flow in the polar vortex.")

*Lines 322-323: Is this because the vortex has weakened (post SPV maximum) to an extent that it is more receptive to upward-propagating waves from below?*

This point is discussed in context with Fig. 11 in lines 388-396. We presume that this behavior is indeed connected to a deceleration of the polar vortex, even though the specific Rossby wave signature differs between pre-SSW and post-SPV.

*Grammatical Comments:*

*Lines 29-32: I would rewrite this opening sentence. It is a little confusing to read and disjointed (especially the part before the colon). The part after the colon I would just write as a new sentence.*

**Thanks for highlighting this sentence. We modified in lines 29-32 to: "While tropospheric and stratospheric Rossby waves can actively affect the state of the stratospheric background flow, their propagation is also affected by it. The structure and strength of the three-dimensional background flow in which the waves propagate can cause them to break (e.g., in case of easterly flow; Charney and Drazin, 1961) or waves can be reflected back towards the surface (Harnik and Lindzen, 2001).")**

*Lines 40-43: The opening sentence is again a bit disjointed. The NAM and AO you put in the first set of parentheses are zonally-averaged themselves, whereas you add about zonally-averaged quantities directly after, and using Hall et al. (2021) as an example. The Baldwin and Dunkerton 2001 and Thompson et al 2006 papers also use zonally-averaged quantities. Multiple sets of parentheses (in this case, 3!) can be confusing and break up the sentence too much. I suggest:*

*"The impact of SSW and SPV events on the tropospheric circulation is most often analyzed in terms of circulation indices of zonally averaged quantities (such as the Arctic Oscillation or the Northern Annular Mode, e.g., Baldwin and Dunkerton, 2001; Thompson et al., 2006), or in terms of changes in the frequency of weather regimes (e.g., Charlton-Perez et al., 2018; Domeisen et al., 2020c; Hall et al., 2023)."*

**We agree. Your suggestion improves the readability of that sentence. (Edit sentence: "The impact of SSW and SPV events on the tropospheric circulation is most often analyzed in terms of circulation indices or zonally averaged quantities (such as the Arctic Oscillation or the Northern Annular Mode, e.g., Baldwin and Dunkerton, 2001; Thompson et al., 2006; Hall et al., 2021), or in terms of changes in the frequency of weather regimes (e.g., Charlton-Perez et al., 2018; Domeisen et al., 2020c; Hall et al., 2023)**

*Line 121: update end of line to: '...allow ONE to neglect...'*

**Thanks! (line 126-127: "...allow to neglect..." → "...allow ONE to neglect...")**

*Line 214: change to '... a value twice as large. '*

**Well spotted. (Edit line 220: "... a twice as high value." → "... a value twice as large.")**

*Line 359: 'reflect' –> 'lead to' or 'result in'*

**Good point. (Change in line 376: "reflect" → "result in")**

*References:*

*Hitchcock, P., and Haynes, P. (2016), Stratospheric control of planetary waves, Geophys. Res. Lett., 43, 11,884–11,892, doi:10.1002/2016GL071372.*

*Hitchcock, P., and I. R. Simpson, 2016: Quantifying Eddy Feedbacks and Forcings in the Tropospheric Response to Stratospheric Sudden Warmings. J. Atmos. Sci., 73, 3641–3657, https://doi.org/10.1175/JAS-D-16-0056.1.*

*Matsuno, T. (1971), A dynamical model of the stratospheric sudden warming,J. Atmos. Sci.,28, 1479–1494.*

*White, I. P., C. I. Garfinkel, and P. Hitchcock, 2022: On the Tropospheric Response to Transient Stratospheric Momentum Torques. J. Atmos. Sci., 79, 2041–2058, https://doi.org/10.1175/JAS-D-21-0237.1.*

**References:**

Bett, P. E., Scaife, A. A., Hardiman, S. C., Thornton, H. E., Shen, X., Wang, L., and Pang, B.: Using large ensembles to quantify the impact of sudden stratospheric warmings and their precursors on the North Atlantic Oscillation, Weather Clim. Dynam., 4, 213–228, https://doi.org/10.5194/wcd-4-213-2023, 2023.

Butler, A.: Table of major mid-winter SSWs in reanalyses products, https://csl.noaa.gov/groups/csl8/sswcompendium/majorevents.html, 2020.

Charlton, A. J. and Polvani, L. M.: A New Look at Stratospheric Sudden Warmings. Part I: Climatology and Modeling Benchmarks, J. Climate, 20, 449–469, https://doi.org/10.1175/JCLI3996.1, 2007a.

Charlton-Perez, A. J., Ferranti, L., and Lee, R. W.: The influence of the stratospheric state on North Atlantic weather regimes, Q J Roy. Meteor. Soc., 144, 1140–1151, https://doi.org/10.1002/qj.3280, 2018.

Charlton-Perez, A. J., Ferranti, L., and Lee, R. W.: The influence of the stratospheric state on North Atlantic weather regimes, Q J Roy. Meteor. Soc., 144, 1140–1151, https://doi.org/10.1002/qj.3280, 2018.

Kolstad, E. W., Lee, S. H., Butler, A. H., Domeisen, D. I., and Wulff, C. O.: Diverse surface signatures of stratospheric polar vortex anomalies, J. Geophys. Res.-Atmos., 127, e2022JD037 422, 2022.

Oehrlein, J., Chiodo, G., and Polvani, L. M.: The effect of interactive ozone chemistry on weak and strong stratospheric polar vortex events, Atmos. Chem. Phys., 20, 10 531–10 544, https://doi.org/10.5194/acp-20-10531-2020, 2020.

Scaife, A. A., Karpechko, A. Y., Baldwin, M. P., Brookshaw, A., Butler, A. H., Eade, R., Gordon, M., MacLachlan, C., Martin, N., Dunstone, N., and Smith, D.: Seasonal winter forecasts and the stratosphere, Atmos. Sci. Lett., 17, 51–56, https://doi.org/10.1002/asl.598, 2016.

Smith, K. L., Polvani, L. M., and Tremblay, L. B.: The Impact of Stratospheric Circulation Extremes on Minimum Arctic Sea Ice Extent, J. Climate, 31, 7169–7183, https://doi.org/10.1175/JCLI-D-17-0495.1, 2018.

Spaeth, J. and Birner, T.: Stratospheric modulation of Arctic Oscillation extremes as represented by extended-range ensemble forecasts, Weather Clim. Dynam., 3, 883–903, https://doi.org/10.5194/wcd-3-883-2022, 2022.

White, I. P., C. I. Garfinkel, and P. Hitchcock: On the Tropospheric Response to Transient Stratospheric Momentum Torques. J. Atmos. Sci., 79, 2041–2058, https://doi.org/10.1175/JAS-D-21-0237.1, 2022.

---

## Author Response (AR2)

**Response to comments by the co-editor**

Michael Schutte, Jacopo Riboldi, Daniela Domeisen

Uppsala, February 28, 2024

*Thank you for your detailed revision and reply to the referees' comments. Based on the reviews and my own evaluation, the manuscript can be acceptable for publication after some minor revision. Here are my suggestions for the paper revision.*

**We would like to thank the co-editor for handling the manuscript and for the following feedback. This has helped us greatly to highlight main messages and to clarify several points more precisely.**

*1. Most of the authors' results on the change of spectral power in the stratosphere show large values over the high zonal wavenumbers (i.e. Figs. 3, 5, 9-11). In the response (i.e. the response to Reviewer 1's comment on Line 325), the authors explain that part of this is likely due to the break-up and filamentation of the polar vortex. Besides this, I wonder whether this is also greatly due to the fact that the authors actually plot the percentage of spectral change in these figures. In the stratosphere, I would expect those high zonal wavenumbers with very small spectral power (or standard deviation for co-spectra) in climatology, which is the denominator when calculating the percentage ("relative anomaly"). Therefore, small change of spectral power in those wavenumbers can result in large and significant change in the percentage. If this is the situation, first, I wonder the change in the absolute value of the spectral power in the high zonal wavenumbers. Is it comparable to the change in planetary scale waves? Are most of these high values in synoptic scales with physical meanings? Second, to avoid any misunderstanding, I suggest the authors further clarify this and add some discussion on this in the manuscript. For example, in the figure caption or title, the authors can state directly that the figures actually plot the percentage of change (In the current caption of figures, the meaning of relative anomalies is not clear enough).*

**Thank you so much for highlighting this point. We discussed this internally already a few times and decided in the end to include the relative anomalies in the figures of the manuscript. The 'raw' anomalies showing the absolute magnitude would point out which harmonics have a relevant contribution in reality from an energetic perspective. Thus, only parts where anomalies of spectral power are high in total would be visible in the figures. This would include mostly harmonics within the black contours that show the NDJFM seasonal mean (Fig. 1).**

**In contrast, the relative anomalies that depict the percentage change highlight rather the signal that one can detect with respect to the variability of spectral power at each harmonic. For example, SSWs and SPVs exhibit a statistically**

[Figure]

**Figure 1.** Anomalies of spectral power $S_{\overline{V'V'}}$ at $10\,\text{hPa}$ averaged over 10-day time intervals around SSW events (shaded). Subplot (a) shows spectra for the period from 10 days prior to the event to the event start, subplot (b) shows spectra between the event start and 10 days after the event start, (c) for 10 to 20 days, and (d) for 20 to 30 days after the event. Wavenumber-phase speed pairs marked with an $\times$ exceed the 0.5th or 99.5th percentile of the re-sampled distribution. Black contour lines show the NDJFM climatology ranging from $0.2\,\text{m}^2\,\text{s}^{-2}\,\Delta\text{c}^{-1}$ to $1.8\,\text{m}^2\,\text{s}^{-2}\,\Delta\text{c}^{-1}$ in steps of $0.2\,\text{m}^2\,\text{s}^{-2}\,\Delta\text{c}^{-1}$.

significant signal from their mean for westward-propagating Rossby waves in the range outside the climatologically relevant harmonics already the days preceding the events. Even though the anomalies might play only a minor role in the overall energy contribution, we decided to highlight this in our manuscript, to point out the importance of anomalies being extremely high or low compared to their climatological mean. Furthermore, since the anomalies are still significant and are part of a coherent pattern across the whole range of harmonics, they are likely also part of the overall Rossby wave behavior during SSWs and SPVs, e.g., the filamentation that we suspect to be represented by higher zonal wavenumbers in the stratospheric spectra during the onset of SSWs.

We followed your suggestion and clarified the captions of figures 3-6. The difference between percentage change and the actual change of spectral power is discussed in the very last paragraph of the outlook. We extended that discussion with respect to your comment (line 452-454).

*2. In the first round of review, both reviewers comment on the use of 48 m/s to define the strong polar vortex (SPV) events and suggest checking the results by using the one standard deviation to define the strong event. I appreciate that the authors tried the later definition in the response letter and showed that the results are not very sensitive to the detailed definition. Though there is no need to add this test in the manuscript, it might be better to mention the result in section 2.1 when introducing the SPV definition, saying that the results are still hold if using an alternative definition.*

Good point, thank you. We have added a sentence in section 2.1 to highlight that results are still valid with the alternative definition using one standard deviation (line 90-92).

*3. In the last review, both reviewers hope that the authors use a longer dataset or model simulations to better test the results. Though the authors argue that this could be a topic of future studies, the authors at least can add more discussion on the possible caveat of the manuscript given the limited number of SSW and SPV events in the analysis, and the possibility of testing the results using model simulations in the Outlook part of the paper.*

Thank you, we extended the paragraph in the discussion about the limited number of events and possibilities to overcome this (line 437-444).

*4. Line 139: I'm not sure whether it is appropriate to use "interpolation" to describe the conversion of spectra from the frequency space to the phase-speed space. Using the word "convert" or "transfer" seems more accurate.*

We choose to use the word "interpolation" to be consistent with the relevant piece of literature describing the procedure, i.e., Randel and Held (1991) (cf. the final words of their page 689). The procedure of interpolation is also described in detail in the Supplementary Information of Riboldi et al. (2022). Thus, for clarity and for consistency with previous literature, we believe it would be preferable to keep this wording.

*Thank you for your patience in waiting the decision.*